# On the Representation Power of Set Pooling Networks

**Christian Bueno**
Department of Mathematics
University of California, Santa Barbara
Santa Barbara, CA 93106
christianbueno@ucsb.edu

**Alan G. Hylton**
Space Communications and Navigation
NASA Glenn Research Center
Cleveland, OH 44135, USA
alan.g.hylton@nasa.gov

## Abstract

Point clouds and sets are input data-types which pose unique problems to deep learning. Since sets can have variable cardinality and are unchanged by permutation, the input space for these problems naturally form infinite-dimensional non-Euclidean spaces. Despite these mathematical difficulties, *PointNet* [18] and *Deep Sets* [31] introduced foundational neural network architectures to address these problems. In this paper we present a unified framework to study the expressive power of such networks as well as their extensions beyond point clouds (partially addressing a conjecture on the extendibility of DeepSets along the way). To this end, we demonstrate the crucial role that the Hausdorff and Wasserstein metrics play and prove new cardinality-agnostic universality results to characterize exactly which functions can be approximated by these models. In particular, these results imply that PointNet generally cannot approximate averages of continuous functions over sets (e.g. center-of-mass or higher moments) implying that DeepSets is strictly more expressive than PointNet in the constant cardinality setting. Moreover, we obtain explicit lower-bounds on the approximation error and present a simple method to produce arbitrarily many examples of this failure-mode. Counterintuitively, we also prove that in the unbounded cardinality setting that any function which can be uniformly approximated by both PointNet and normalized-DeepSets must be constant. Finally, we also prove theorems on the Lipschitz properties of PointNet and normalized-DeepSets which shed insight into exploitable inductive bias in these networks.

## 1 Introduction

The architectures introduced in *PointNet* [18] and *Deep Sets* [31] are foundational contributions to the direct analysis of point clouds and sets via deep learning. These works provided two of the earliest such methods and continue to be of theoretical interest since both provide simple architectures which are *permutation-invariant* by construction. In addition to this, a single invariant model of either approach can handle sets of differing sizes and can theoretically scale to any cardinality with enough computational resources.

In either case the procedure to process a point cloud $A$ in $\mathbb{R}^n$ is simply as follows: (1) apply a network $\varphi$ to each element of $A$, (2) apply a permutation-invariant pooling operation to aggregate these point-features into a global feature for $A$ (i.e. max-pooling for PointNet, sum-pooling for DeepSets), and lastly (3) pass this global feature to the second network $\rho$ to obtain the final output. For general PointNet and DeepSets permutation-invariant models, this can be concisely expressed mathematically as

$$\psi_{\text{PointNet}}(A) = \rho\left(\max_{a \in A} \varphi(a)\right) \quad \text{and} \quad \psi_{\text{DeepSets}}(A) = \rho\left(\sum_{a \in A} \varphi(a)\right),$$

35th Conference on Neural Information Processing Systems (NeurIPS 2021).

respectively. Depending on the output layer of $\rho$ this can be used for either set-regression or set-classification. Note that arbitrarily large point clouds can be passed through and that the permutation-invariance of the max/sum-pooling operations ensure that rearrangement of the elements of $A$ do not alter the output.

Importantly, each of these prior works provide their own universal approximation theorems (UATs) to support the empirical success of their methods. In this work we prove substantial extensions of these results motivated by the following points:

**Unified Approach and Minimal Universality:** The pre-existing universality results use very different approaches. Additionally, these results do not explore the minimum architecture requirements needed for universality. In this paper we present a proof method that applies to more than one model and provides universal "shallow" examples.

**Cardinality Limitations:** The universality results in [18, 31] both assume that the cardinality of the inputs are fixed to some size $n$ and thus do not make use of the cardinality-agnostic nature of these models. This gap is of worthy of investigation because (1) real-world point cloud datasets can have heterogeneous cloud sizes and (2) it may happen that the deployed model encounters point clouds with cardinalities that did not exist in the training data (e.g. if better sensors become available). It is not immediately clear whether one should expect these model classes to have enough power to universally approximate the functions of interest when allowing for such changes in set size.

**Infinite Input-Layer Width Limit:** One approach for learning with mesh inputs is to sample the mesh and feed the resulting point cloud into a suitable neural network [18, 19, 9]. Although PointNet and DeepSets can readily accept samples of any size and via any sampling method, the computation graph at the input layers necessarily widens as the point clouds get larger. Understanding the expressiveness and consistency as sampling cardinality (and hence input-layer size) grows to infinity may provide theoretical insight on this approach to learning from meshes. Models such as Neural ODEs [6] and Neural Tangent Kernels [11] have benefited from similar considerations (infinite depth and infinite hidden-layer width limits respectively).

**DeepSets Extension Conjecture:** It was conjectured in the supplementary materials of [31] (below their proof Theorem 9) that the DeepSets invariant model should be extendable to input sets of countably infinite or even uncountably infinite size and retain universality in that setting. However, they note that there are fundamental topological obstructions to answering this question. We provide partial resolutions to this conjecture by showing that a change in the pooling function allows for inputs beyond finite sets (Sec. 3.4).

**Comparison of Representational Power:** In this work we slightly modify the permutation-invariant DeepSets model by use of average-pooling instead of sum-pooling. The only change is re-scaling by the set size before applying $\rho$ so we call this version **normalized-DeepSets**. We find that:

- PointNet (normalized-DeepSets) can uniformly approximate only the functions which are uniformly continuous with respect to the Hausdorff (Wasserstein) metric (Theorem 3.4).
- Only the constant functions can be simultaneously uniformly approximated by both PointNet and normalized-DeepSets when input sets are allowed to be arbitrarily large (Theorem 4.1).
- Even when cardinality is fixed to size $k$ there is a substantial difference in approximation power. In particular, PointNet cannot uniformly approximate averages of continuous functions over sets such as the center-of-mass or higher moments for $k \geq 3$. We prove an explicit error lower bound for learnability of these functions by PointNet and provide a simple method for generating examples of problematic inputs for this failure-mode (Theorem 4.2).

**Lipschitz Properties and Inductive Bias.** In this work we also show that PointNet and normalized-DeepSets are Lipschitz with respect to the Hausdorff and 1-Wasserstein metrics (Theorem 3.7). This result can be seen as a quantitative generalization of the qualitative stability theorem in [18]. Moreover, the Lipschitz-ness of these models suggest an interpretable inductive bias which we exploit in the experiments in Sec. 5.

**Related work.** Besides permutation-invariant networks, there also exists permutation-*equivariant* networks which have the property that a permutation of the inputs results in a corresponding permutation of the output. In addition to the invariant PointNet model in [18] and the invariant DeepSets model in [31], equivariant models are also introduced. Though these equivariant networks are also

commonly referred to as PointNet and DeepSets networks, they are fundamentally different and are in general *not* permutation-invariant. In this paper, we will only consider the invariant models.

After the introduction of PointNet and DeepSets, other deep learning approaches for sets and point clouds quickly appeared. These include hierarchical variants such as PointNet++ [19] and attention-based models such as Set Transformer [13]. Models which were additionally equivariant to 3D rigid motions were also developed such as the hierarchical Tensor Field Networks [24] and the attention-based SE(3)-Transformer [8]. There is also the closely related field of graph neural networks (GNNs) which shares many similar difficulties and questions as do neural networks for sets. See [9] and [27] for comprehensive surveys on deep learning methods for point clouds and graphs respectively.

Whether a permutation-invariant function $F$ can be *exactly* represented by sum-decomposition – i.e. whether $F(A) = \rho(\sum_{a \in A} \varphi(a))$ for some $\rho$ and $\varphi$ – has been addressed by [31, 26, 28] with positive and negative results depending on whether the point clouds come from a countable or uncountable universe. In particular, [31] proves that sum-decomposition is possible when the sets are drawn from a countable universe and [26] shows that there are continuous permutation-invariant functions that are not sum-decomposable when the latent space is too small (and similarly for max-decomposition).

Approximation results for other permutation-equivariant/invariant models have also been presented in the literature. In [22] and [15] special kinds of equivariant models are introduced and proofs of universality are provided. In [29] polynomial-based $G$-invariant networks are introduced for which a fixed-cardinality universal approximation results are proved via the Stone-Weierstrass theorem and Hilbert's finiteness theorems from classical invariant theory.

Lastly, there has been much closely related work on the expressiveness of GNNs. GNN universality results are proved in [12, 1], the degree to which GNNs can generalize to graphs of varying sizes is investigated in [30], and the relative power of max/mean/sum-pooling in GNNs is studied in [28].

## 2    Preliminaries

Our focus will be on understanding which functions can be uniformly approximated by architectures like that of *PointNet* and *Deep Sets*. Although tasks such as point cloud segmentation or point cloud generation are not directly addressed, some results may apply depending on architecture details. All presented theorems have proofs in the paper's main body or in the supplementary materials.

### 2.1    Set Pooling Networks: PointNet, DeepSets, & Normalized-DeepSets

In addition to the max-pooling of PointNet and sum-pooling of DeepSets, we also consider average-pooling. For $A \subseteq \mathbb{R}^n$ a finite point cloud, the network architectures are respectively given by

$$\psi_{\max}(A) = \rho\left(\max_{a \in A} \varphi(a)\right), \quad \psi_{\mathrm{ave}}(A) = \rho\left(\frac{1}{|A|} \sum_{a \in A} \varphi(a)\right), \quad \psi_{\mathrm{sum}}(A) = \rho\left(\sum_{a \in A} \varphi(a)\right).$$

Here $\varphi : \mathbb{R}^n \to \mathbb{R}^m$ creates features for each point in $A$ which gets symmetrically pooled and then passed to $\rho : \mathbb{R}^m \to \mathbb{R}^\ell$ (where $\max$ is the component-wise maximum). For simplicity, we will usually take $\ell = 1$ and in practice we want both $\rho$ and $\varphi$ to be neural networks or some other parametrized model with learnable parameters. Once again, note that because of the pooling operation before $\rho$, the output will not depend on the ordering of points in the point cloud, and the size of the point cloud is not an issue because the pooling operations scale to arbitrary finite cardinalities. In general, we call these and any other network of the form $\rho(\mathrm{pool}_{a \in A} \varphi(a))$ a **set pooling network**.

It may at first seem that the use of average-pooling instead of sum-pooling makes no difference. Indeed, this is true when cloud cardinality is fixed since then the $|A|^{-1}$ factor is constant and can be absorbed into $\rho$. However, when cardinality is free to change this factor can no longer be absorbed and makes a difference. To see this, imagine a finer and finer uniform sampling of the unit circle with $\varphi > 0$ everywhere. If we use average-pooling, the inside of $\rho$ will converge to the average value of $\varphi$ on the circle, but will explode if we use sum-pooling. This good behavior in the large cardinality limit will be critical later on.

It will be useful to introduce some abstractions and simplifying notation. Let $\mathcal{F}(\Omega)$ denote the set of all nonempty finite subsets of a set $\Omega$ (i.e. point clouds in $\Omega$), $\mathcal{F}^{\leq k}(\Omega)$ the set of nonempty subsets of

size $\leq k$, and $\mathcal{F}^k(\Omega)$ the set of $k$-point subsets. Now consider a set $\Omega$, a function $\boldsymbol{f} : \Omega \to \mathbb{R}^n$, and define $\max_{\boldsymbol{f}}, \mathrm{ave}_{\boldsymbol{f}}, \mathrm{sum}_{\boldsymbol{f}} : \mathcal{F}(\Omega) \to \mathbb{R}^n$ as the set-functions given by component-wise pooling

$$\max\nolimits_{\boldsymbol{f}}(A) = \max_{\boldsymbol{a} \in A} \boldsymbol{f}(\boldsymbol{a}), \quad \mathrm{ave}_{\boldsymbol{f}}(A) = \frac{1}{|A|} \sum_{\boldsymbol{a} \in A} \boldsymbol{f}(\boldsymbol{a}), \quad \mathrm{sum}_{\boldsymbol{f}}(A) = \sum_{\boldsymbol{a} \in A} \boldsymbol{f}(\boldsymbol{a}).$$

With these conventions, $\rho \circ \max_{\varphi}$, $\rho \circ \mathrm{sum}_{\varphi}$, and $\rho \circ \mathrm{ave}_{\varphi}$ will be the general form of what we call PointNet, DeepSets, and normalized-DeepSets architectures respectively. The case of $\rho \circ \mathrm{sum}_{\varphi}$ will prove difficult to treat in the same manner as the others and so will get a separate treatment (Sec. 3.4).

## 2.2 Function Spaces and Uniform Approximation

Here we review some standard functional analysis. For more detailed background see [21, 20, 3].

Let $\mathcal{C}(X, X')$ and $\mathcal{U}(M, M')$ denote respectively the set of all continuous functions from topological space $X$ to topological space $X'$ and the set of all *uniformly* continuous functions from metric space $(M, d)$ to metric space $(M', d')$. Unless otherwise specified we always assume that $\mathbb{R}^n$ carries the standard Euclidean norm $\|\cdot\|$ and for simplicity we let $\mathcal{C}(X) := C(X, \mathbb{R})$ and $\mathcal{U}(M) := \mathcal{U}(M, \mathbb{R})$.

For any set $A$, we define $\mathcal{B}(A)$ to be the set of all $\mathbb{R}$-valued bounded functions on $A$. With this we can define the bounded function spaces $\mathcal{C}_b(X) = \mathcal{C}(X) \cap \mathcal{B}(X)$ and $\mathcal{U}_b(M) = \mathcal{U}(M) \cap \mathcal{B}(M)$. When $X$ is compact or $M$ is precompact (i.e. has compact metric completion) then we have $\mathcal{C}(X) = \mathcal{C}_b(X)$ and $\mathcal{U}(M) = \mathcal{U}_b(M)$. We can endow all of these $\mathbb{R}$-valued functions spaces with the *topology of uniform convergence* by equipping them with the supremum norm[1] given by $\|\!|f|\!\|_A = \sup_{a \in A} |f(a)|$. In particular, the bounded families $\mathcal{U}_b(M), \mathcal{C}_b(X), \mathcal{B}(A)$ are all Banach spaces with this norm.

The *uniform closure* of a family of $\mathbb{R}$-valued functions $\mathcal{A}$ is the set of all functions which can be uniformly approximated via members of $\mathcal{A}$, i.e. the topological closure with respect to $\|\!|\cdot|\!\|$. One of our key focuses will be determining the uniform closure of families of set pooling networks as this tells us exactly which functions can and can't be approximated.

Lastly, given an injective map $i : A \to X$, we say that $\varphi : A \to \mathbb{R}$ (uniquely) continuously extends to $X$ if there is a (unique) $\widetilde{\varphi} \in \mathcal{C}(X)$ such that $\widetilde{\varphi} \circ i = \varphi$. We say a family of functions $\mathcal{N}$ on $A$ (uniquely) continuously extends to $X$ if every $\varphi \in \mathcal{N}$ (uniquely) continuously extends to $X$. One nice property of uniformly continuous functions on a metric space $(M, d)$ is that they uniquely continuously extend to the metric completion $(\overline{M}, \overline{d})$ i.e. $\mathcal{U}(M)$ uniquely continuously extends to $\overline{M}$.

## 2.3 Topologies and Metrics for Point Clouds

From now on we will assume $(\Omega, d)$ is a compact metric space unless otherwise stated and when $\Omega \subseteq \mathbb{R}^n$ it will be a compact set equipped with the Euclidean metric (e.g. $[0, 1]^n$). Let $\mathcal{K}(\Omega)$ denote the set of all compact subsets of $\Omega$ and $\mathcal{P}(\Omega)$ denote the set of all Borel probability measures on $\Omega$. The **Hausdorff metric** $d_H$ [17, 16] is a natural metric for $\mathcal{K}(\Omega)$ and 1-**Wasserstein metric** $d_W$ [25] (also called the Earth-mover distance) is a natural metric for $\mathcal{P}(\Omega)$. With these metrics, $\mathcal{K}(\Omega)$ and $\mathcal{P}(\Omega)$ become *compact* metric spaces of their own whenever $(\Omega, d)$ is compact [10, 25]. From now on we will assume these two spaces are always equipped with the aforementioned metrics.[2]

We also briefly mention $M(\Omega)$, the Banach space of finite signed regular Borel measures on $\Omega$. By the Riesz-Markov theorem it is the topological dual space of $\mathcal{C}(\Omega)$. Of interest to us is that $\mathcal{P}(\Omega) \subseteq M(\Omega)$ and that the weak-* topology on $\mathcal{P}(\Omega)$ coincides with the topology induced by $d_W$. This means that $d_W(\mu_n, \mu) \to 0$ iff $\int f \, d\mu_n \to \int f \, d\mu$ for all $f \in \mathcal{C}(\Omega)$.

Next, note that $\mathcal{F}(\Omega) \subseteq \mathcal{K}(\Omega)$ and let $i_{\mathcal{K}}$ denote the natural inclusion map. We can also define an injective map $i_{\mathcal{P}} : \mathcal{F}(\Omega) \to \mathcal{P}(\Omega)$ by mapping $A \in \mathcal{F}(\Omega)$ to its associated empirical measure $i_{\mathcal{P}}(A) = \mu_A = \frac{1}{|A|} \sum_{a \in A} \delta_a \in \mathcal{P}(\Omega)$ where $\delta_a$ is the Dirac delta measure supported at $a$. The injective maps $i_{\mathcal{K}}$ and $i_{\mathcal{P}}$ allow us to induce the $d_H$ and $d_W$ metrics on $\mathcal{F}(\Omega)$. We will denote the metrized versions by $\mathcal{F}_H(\Omega)$ and $\mathcal{F}_W(\Omega)$ respectively and use the same convention for the bounded

---

[1]In general, $\|\!|\cdot|\!\|$ will not always be a true norm on $\mathcal{C}(X)$ or $\mathcal{U}(M)$ since there may unbounded functions.

[2]The topology induced on $\mathcal{K}(\Omega)$ and $\mathcal{P}(\Omega)$ by $d_H$ and $d_W$ depends only on the topology of $(\Omega, d)$ and not on the choice of metric $d$. These topologies are called the Vietoris [16] and weak-* topologies [21] respectively. Indeed, in parts of this paper we could have proceeded without metrics at all and directly used these topologies. However, the particular metrics $d_h$ and $d_W$ will be necessary to prove certain results such as those in Sec. 3.3.

cardinality spaces e.g. $\mathcal{F}_H^k(\Omega)$ and $\mathcal{F}_W^k(\Omega)$. Importantly, $i_{\mathcal{K}}$ and $i_{\mathcal{P}}$ embed $\mathcal{F}(\Omega)$ as a *dense* subset of $\mathcal{K}(\Omega)$ and $\mathcal{P}(\Omega)$ respectively. The former follows from compactness of the members of $\mathcal{K}(\Omega)$ and the latter is true because you can approximate any distribution with samples (see [7, 25] for details).

For $f \in \mathcal{C}(\Omega)$, define $\text{Max}_f : \mathcal{K}(\Omega) \to \mathbb{R}$ and $\text{Ave}_f : \mathcal{P}(\Omega) \to \mathbb{R}$ as the functions given by $\text{Max}_f(K) = \max_{x \in K} f(x)$ and $\text{Ave}_f(\mu) = \int_\Omega f \, d\mu = \mathbb{E}_{x \sim \mu}[f(x)]$.

**Lemma 2.1.** *Let $(\Omega, d)$ be compact, $f \in \mathcal{C}(\Omega)$. Then $\text{Max}_f \in \mathcal{C}(\mathcal{K}(\Omega))$ and $\text{Ave}_f \in \mathcal{C}(\mathcal{P}(\Omega))$ and $\text{Max}_f \circ i_{\mathcal{K}} = \max_f$ and $\text{Ave}_f \circ i_{\mathcal{P}} = \text{ave}_f$. As a consequence, PointNet and normalized-DeepSets are uniformly continuous on $\mathcal{F}_H(\Omega)$ and $\mathcal{F}_W(\Omega)$ respectively.*

This lemma tells us that $\max_f$ and $\text{ave}_f$ continuously extend to $\mathcal{K}(\Omega)$ and $\mathcal{P}(\Omega)$ and hence PointNet and normalized-DeepSets must as well (since composing with the continuous $\rho$ preserves continuity). Thus, we will be able analyze such architectures as continuous functions on compact metric spaces, which is mathematically a much better setting than set-theoretic functions on an un-metrized $\mathcal{F}(\Omega)$.

### 2.4 Notation for Families of Neural Networks

In this section we will introduce notation to help abstract away neural network architecture details and instead focus on classes of neural networks and the functions they can represent.

For a collection $\mathcal{A}$ of functions from $X$ to $Y$ and a collection $\mathcal{B}$ of functions from $Y$ to $Z$, we denote the set of all compositions by $\mathcal{B} \circ \mathcal{A} = \{f \circ g \mid f \in \mathcal{B}, g \in \mathcal{A}\}$. In the case of a single function $\sigma : Y \to Z$ we let $\sigma \circ \mathcal{A} = \{\sigma\} \circ \mathcal{A} = \{\sigma \circ f \mid f \in \mathcal{A}\}$. For a collection of $\mathbb{R}$-valued functions $\mathcal{A}$, span $\mathcal{A}$ is the set of all linear combinations of the functions in $\mathcal{A}$.

Next, let $\mathcal{N}^\sigma$ denote the set of all functions which can be written as $f(\boldsymbol{x}) = \sum_{i=1}^n a_i \sigma(\boldsymbol{w}_i \cdot \boldsymbol{x} + b_i)$, i.e. the set of single-hidden layer neural networks with activation $\sigma$ and linear scalar output-layer. Deeper $H$-layered network classes are denoted by $\mathcal{N}^{\boldsymbol{\sigma}}$ where $\boldsymbol{\sigma} = (\sigma_1, \ldots, \sigma_H)$ is a list of $H$-many activation functions which can be deepened like so $\mathcal{N}^{\boldsymbol{\sigma}, \tau} := \mathcal{N}^{\sigma_1, \ldots, \sigma_H, \tau} := \text{span}(\tau \circ \mathcal{N}^{\boldsymbol{\sigma}})$.

Next we define various classes of PointNets. Let $\mathcal{N}_{\max}^{\boldsymbol{\sigma}} := \text{span}\{\max_f \mid f \in \mathcal{N}^{\boldsymbol{\sigma}}\}$ then define $\mathcal{N}_{\max}^{\boldsymbol{\sigma}|\tau} := \text{span}(\tau \circ \mathcal{N}_{\max}^{\boldsymbol{\sigma}})$. We use the subscript 'max' to recall which pooling is used and to remind ourselves that these networks take *set-inputs*, not vector-inputs. As before, we can inductively define deeper networks and denote them via vector-bold activations as in $\mathcal{N}_{\max}^{\boldsymbol{\sigma}|\boldsymbol{\tau}}$, but we now also distinguish by the placement of '|' whether the depth occurs before or after the pooling.

Next we do the same for normalized-DeepSets. Let $\mathcal{N}_{\text{ave}}^{\boldsymbol{\sigma}} := \text{span}\{\text{ave}_f \mid f \in \mathcal{N}^{\boldsymbol{\sigma}}\}$. Note that $\text{ave}_f + \text{ave}_g = \text{ave}_{f+g}$ and $\alpha \text{ave}_f = \text{ave}_{\alpha f}$. Thus since $\mathcal{N}^{\boldsymbol{\sigma}}$ is a linear space, taking the span has no effect and so $\mathcal{N}_{\text{ave}}^{\boldsymbol{\sigma}} = \{\text{ave}_f \mid f \in \mathcal{N}^{\boldsymbol{\sigma}}\}$. Adding a layer post-pooling yields the new family $\mathcal{N}_{\text{ave}}^{\boldsymbol{\sigma}|\tau} := \text{span}(\tau \circ \mathcal{N}_{\text{ave}}^{\boldsymbol{\sigma}})$ which gets us new functions. Like with PointNet, we can inductively develop deeper families by deepening pre-pooling or post-pooling.

By Lemma 2.1 we can extend all the operations of our neural networks to $\mathcal{K}(\Omega)$ and $\mathcal{P}(\Omega)$ in a natural way. This let's us talk about about PointNet networks on $\mathcal{K}(\Omega)$ and normalized-DeepSets networks on $\mathcal{P}(\Omega)$. We define them analogously by replacing $\max_f$ with $\text{Max}_f$ and $\text{ave}_f$ with $\text{Ave}_f$. Thus

$$\mathcal{N}_{\text{Max}}^{\boldsymbol{\sigma}|\tau} = \text{span}(\tau \circ \mathcal{N}_{\text{Max}}^{\boldsymbol{\sigma}}), \qquad \mathcal{N}_{\text{Ave}}^{\boldsymbol{\sigma}|\tau} = \text{span}(\tau \circ \mathcal{N}_{\text{Ave}}^{\boldsymbol{\sigma}})$$

where $\mathcal{N}_{\text{Max}}^{\boldsymbol{\sigma}} = \text{span}\{\text{Max}_f \mid f \in \mathcal{N}^{\boldsymbol{\sigma}}\}$ and $\mathcal{N}_{\text{Ave}}^{\boldsymbol{\sigma}} = \text{span}\{\text{Ave}_f \mid f \in \mathcal{N}^{\boldsymbol{\sigma}}\}$. As before, the linear structure of $\mathcal{N}^{\boldsymbol{\sigma}}$ means $\mathcal{N}_{\text{Ave}}^{\boldsymbol{\sigma}}$ could have been defined without the span. Here the pooling subscripts matter much more as a 'Max' subscript implies that the networks take compact sets as inputs and an 'Ave' subscript implies that the networks take probability measures as inputs.

For further explanation of this notation and examples it describes see Appendix C.

## 3 Universality Results

### 3.1 Topological UAT

In [14] it is proven that $\mathcal{N}^\sigma$ with $\sigma \in \mathcal{C}(\mathbb{R})$ has the universal approximation property iff $\sigma$ is not a polynomial. For this reason, we will say a $\sigma \in \mathcal{C}(\mathbb{R})$ is 'universal' if it is non-polynomial and denote the set of all such such functions by $\mathfrak{U}(\mathbb{R})$. Using this theorem and Stone-Weierstrass we prove a UAT

for certain kinds of generalized neural networks on an abstract compact Hausdorff space. Though we independently arrived at this theorem, we later discovered this result was essentially proven in [23] (Theorem 5.1) for a different context.[3] However, our proof is slightly different and so we provide a detailed proof in the supplementary materials for completeness and the benefit of the reader.

Recall that a family of functions $S$ on $\Omega$ separates points if for any $x \neq y$ there is an $f \in S$ so that $f(x) \neq f(y)$.

**Theorem 3.1** (Topological-UAT). *Let $X$ be a compact Hausdorff space and $\sigma \in \mathfrak{U}(\mathbb{R})$. If $S \subseteq \mathcal{C}(X)$ separates points and contains a nonzero constant, then $\mathrm{span}(\sigma \circ \mathrm{span}\, S)$ is dense in $\mathcal{C}(X)$. Additionally, if $S$ also happens to be a linear subspace, then $\mathrm{span}(\sigma \circ S)$ is dense in $\mathcal{C}(X)$.*

## 3.2 Point Cloud UAT

With the following lemma, we will have met all the conditions required to use the topological-UAT on $\mathcal{K}(\Omega)$ and $\mathcal{P}(\Omega)$.

**Lemma 3.2** (Separation Lemma). *Let $\Omega \subseteq \mathbb{R}^N$ be compact and $\sigma \in \mathfrak{U}(\mathbb{R})$. Then the set of functions $S_{\mathrm{Max}} = \{\mathrm{Max}_f \mid f \in \mathcal{N}^\sigma\}$ and $S_{\mathrm{Ave}} = \{\mathrm{Ave}_f \mid f \in \mathcal{N}^\sigma\}$ separate points and contain constants.*

The following theorems show that one hidden layer in the inner network $\varphi$ and one hidden layer in the outer network $\rho$ suffice to prove the universal approximation theorems for PointNet and normalized-DeepSets (for further explanation see Appendix C).

**Theorem 3.3.** *Let $\Omega \subseteq \mathbb{R}^N$ be compact and $\sigma, \tau \in \mathfrak{U}(\mathbb{R})$. Then $\mathcal{N}_{\mathrm{Max}}^{\sigma|\tau}$ and $\mathcal{N}_{\mathrm{Ave}}^{\sigma|\tau}$ are dense in $\mathcal{C}(A)$ and $\mathcal{C}(B)$ respectively, whenever $A \subseteq \mathcal{K}(\Omega)$ and $B \subseteq \mathcal{P}(\Omega)$ are closed subsets.*

*Proof.* Recall $\mathcal{N}_{\mathrm{Max}}^{\sigma|\tau} = \mathrm{span}\{\tau \circ \mathrm{span}\, S_{\mathrm{Max}}\}$ and $\mathcal{N}_{\mathrm{Ave}}^{\sigma|\tau} = \mathrm{span}\{\tau \circ S_{\mathrm{Ave}}\}$. Since $\mathcal{K}(\Omega)$ and $\mathcal{P}(\Omega)$ are compact metric spaces, and $A$ and $B$ are closed, they must be also be compact Hausdorff. By Lemma 3.2 we know $S_{\mathrm{Max}}$ and $S_{\mathrm{Ave}}$ separate points and contain nonzero constants and so the topological-UAT (Theorem 3.1) yields the desired result. □

Now we restrict this result to $\mathcal{F}(\Omega)$. For $f : \mathcal{F}(\Omega) \to \mathbb{R}$ the following result tells us that under mild hypothesis that (1) PointNets can uniformly approximate $f$ iff $f$ is $d_H$-uniformly continuous and (2) normalized-DeepSets can uniformly approximate $f$ iff $f$ is $d_W$-uniformly continuous.

**Theorem 3.4** (Point-Cloud-UAT). *Let $\Omega \subseteq \mathbb{R}^N$ be compact. If $\sigma, \tau \in \mathfrak{U}(\mathbb{R})$, then the uniform closure of $\mathcal{N}_{\mathrm{max}}^{\sigma|\tau}$ and $\mathcal{N}_{\mathrm{ave}}^{\sigma|\tau}$ within $\mathcal{B}(\mathcal{F}(\Omega))$ is $\mathcal{U}(\mathcal{F}_H(\Omega))$ and $\mathcal{U}(\mathcal{F}_W(\Omega))$ respectively.*

*Proof.* $\mathcal{F}_H(\Omega)$ and $\mathcal{F}_W(\Omega)$ are isometrically isomorphic to $i_{\mathcal{K}}(\mathcal{F}(\Omega))$ and $i_{\mathcal{P}}(\mathcal{F}(\Omega))$ which are in turn dense in $(\mathcal{K}(\Omega), d_H)$ and $(\mathcal{P}(\Omega), d_W)$. By Lemma 2.1 we have that $\mathcal{N}_{\mathrm{max}}^{\sigma|\tau}$ and $\mathcal{N}_{\mathrm{ave}}^{\sigma|\tau}$ continuously extend to $\mathcal{K}(\Omega)$ and $\mathcal{P}(\Omega)$ as $\mathcal{N}_{\mathrm{Max}}^{\sigma|\tau}$ and $\mathcal{N}_{\mathrm{Ave}}^{\sigma|\tau}$. By Theorem 3.3 we know $\mathcal{N}_{\mathrm{Max}}^{\sigma|\tau}$ and $\mathcal{N}_{\mathrm{Ave}}^{\sigma|\tau}$ are dense in $\mathcal{C}(\mathcal{K}(\Omega))$ and $\mathcal{C}(\mathcal{P}(\Omega))$. Finally, by Lemma A.1 we have the desired result. □

It is worth noting that we could have directly used another UAT proven in [23] for neural networks on locally convex spaces [23](Corollary 5.1.2) for the universality of normalized-DeepSets since $\mathcal{P}(\Omega)$ naturally lives within a topological vector space and $\mathrm{Ave}_f$ is always a continuous linear functional. However this result would not directly work for PointNet since $\mathcal{K}(\Omega)$ lacks a natural vector space structure. Hence we chose the above route for consistency of technique and to be self-contained.

We now prove as a corollary a refinement of the universal approximation theorems of [18] and [31], both of which applied to the the case of $k$-point point clouds (for fixed $k$). In this version of the theorem we are able to restrict the depth of the neural network to just three hidden layers. Additionally, as with the above, this result simultaneously establishes which functions *cannot* be uniformly approximated by these architectures. The proof is essentially the same as Theorem 3.4.

**Corollary 3.5.** *Let $\Omega \subseteq \mathbb{R}^N$ be compact. If $\sigma, \tau \in \mathfrak{U}(\mathbb{R})$, then the uniform closure of $\mathcal{N}_{\mathrm{max}}^{\sigma|\tau}$ and $\mathcal{N}_{\mathrm{ave}}^{\sigma|\tau}$ within $\mathcal{B}(\mathcal{F}^k(\Omega))$ are $\mathcal{U}(\mathcal{F}_H^k(\Omega))$ and $\mathcal{U}(\mathcal{F}_W^k(\Omega))$ respectively.*

---

[3] In [23], the motivation was to study expressiveness of neural networks on all of $\mathbb{R}^n$ (not just a compact subset of it) and this necessitated an extension of the classical UAT.

*Proof.* $\mathcal{F}_H^k(\Omega)$ and $\mathcal{F}_W^k(\Omega)$ are isometrically isomorphic to $i_\mathcal{K}(\mathcal{F}^k(\Omega))$ and $i_\mathcal{P}(\mathcal{F}^k(\Omega))$, which are in turn dense in their respective closures which we denote $\mathcal{G}_H(\Omega) \subseteq \mathcal{K}(\Omega)$ and $\mathcal{G}_W(\Omega) \subseteq \mathcal{P}(\Omega)$. Thus by Lemma 2.1 and Theorem 3.3 we have that $\mathcal{N}_{\mathrm{Max}}^{\sigma|\tau}$ and $\mathcal{N}_{\mathrm{Ave}}^{\sigma|\tau}$ are dense in $\mathcal{C}(\mathcal{G}_H(\Omega))$ and $\mathcal{C}(\mathcal{G}_W(\Omega))$. Finally, by Lemma 2.1 we have the desired result. $\square$

### 3.3 Stability and Lipschitz Property

A function $f$ from a metric space $(M_1, d_1)$ to a metric space $(M_2, d_2)$ is Lipschitz if there is a constant $K$ (called the Lipschitz constant) such that $d_2(f(x), f(y)) \leq K d_1(x, y)$ for all $x, y \in M_1$. This is a valuable property for understanding how rapidly $f$ can vary with respect to the inputs, especially in the absence of differentiability. By refining the continuity arguments for $\mathrm{Max}_f$ and $\mathrm{Ave}_f$, we can deduce the Lipschitz constant of these maps from the Lipschitz constant of $f$.

**Lemma 3.6.** *Suppose $(\Omega, d)$ is compact and $f : \Omega \to \mathbb{R}$ is Lipschitz continuous with Lipschitz constant $L$. Then $\mathrm{Max}_f \in \mathcal{K}(\Omega)$ and $\mathrm{Ave}_f \in \mathcal{P}(\Omega)$ have Lipschitz constant $2L$ and $L$.*

As a consequence of Lemma 3.6, we can show that whenever the activation functions are Lipschitz, the whole neural network architecture will be Lipschitz. This is particularly valuable in the variable cardinality case since derivatives with respect to cardinality do not make sense.

**Theorem 3.7.** *Suppose $\Omega \subseteq \mathbb{R}^N$ is compact. Then every PointNet and normalized-DeepSets network with Lipschitz activation functions is Lipschitz on $\mathcal{F}_H(\Omega)$ and $\mathcal{F}_W(\Omega)$ respectively.*

In particular, this means that for a PointNet $F_{\mathrm{max}}$ and a normalized-DeepSets $F_{\mathrm{ave}}$ there exists constants $K_{\mathrm{max}}$ and $K_{\mathrm{ave}}$ so that

$$\|F_{\mathrm{max}}(A) - F_{\mathrm{max}}(B)\| \leq K_{\mathrm{max}} d_H(A, B), \quad \text{and} \quad \|F_{\mathrm{ave}}(A) - F_{\mathrm{ave}}(B)\| \leq K_{\mathrm{ave}} d_H(A, B),$$

where the constants $K_{\mathrm{max}}$ and $K_{\mathrm{ave}}$ are determined by architectures of $F_{\mathrm{max}}$ and $F_{\mathrm{ave}}$ respectively.

One implication of this is that if one were to sample a mesh sufficiently well, and knew the Lipschitz constant of the whole PointNet/normalized-DeepSets network, then it would be possible to bound how big a difference could be produced in the output under two different samplings of the same object. This bound would not depend on the cardinality, but on the quality of the sample with respect to the metrics $d_H$ and $d_W$. This makes Theorem 3.7 something of a quantitative analogue and extension of the stability theorem for PointNet presented in Theorem 2 of [18].

Theorem 3.7 also provides us with a qualitative way to understand what these networks are most responsive to. For example, if two point clouds $A$ and $B$ had very large $d_H(A, B)$ but very small $d_W(A, B)$, it would stand to reason that PointNet would more readily be able to tell them apart than normalized-DeepSets. We explore this promising inductive bias further in Sec. 5.

### 3.4 DeepSets Conjecture and Theoretical Issues with Standard DeepSets

To use the universal approximation results developed here, it was key for the there to exists an underlying compact Hausdorff space on which to study the networks. In the case of PointNet and normalized-DeepSets, the compact Hausdorff space was not obvious, but we identified the appropriate extension spaces $\mathcal{K}(\Omega)$ and $\mathcal{P}(\Omega)$ to which we could unambiguously and continuously extend the networks. This gave a partial answer to the DeepSets extension conjecture of [31] in the sense that if we replace sum-pooling with max-pooling then these networks can be made to theoretically accept sets of uncountably infinite cardinality (so long as they are compact i.e. members of $\mathcal{K}(\Omega)$). Similarly, for sufficiently nice sets like compact smooth manifolds $M \subseteq \mathbb{R}^n$ (e.g. a sphere) one can feed the uniform probability measure $\mu_M$ of $M$ to the extension of normalized-DeepSets to $\mathcal{P}(\Omega)$.

However, we were not able to do this for standard DeepSets. This is because, generally speaking, there is no compact Hausdorff space to which we can continuously extend $\mathrm{sum}_f$ for all reasonable neural networks $f$. This is because a continuous function on a compact space is necessarily a bounded function but $\mathrm{sum}_f : \mathcal{F}(\Omega) \to \mathbb{R}$ is generally unbounded. To illustrate this, let $\Omega = [0, 1] \subseteq \mathbb{R}$ and let $f$ be a continuous function, e.g. a neural network, which is not identically zero. Then there must be a closed interval $K$ on which $|f| > \epsilon$ for some $\epsilon > 0$. From this we can see that $\mathrm{sum}_f(A)$ for $A \subseteq K$ grows arbitrarily large in absolute value as the size of $A$ increases. Thus, in this case, every $\mathrm{sum}_f$ except $\mathrm{sum}_0$ is unbounded on $\mathcal{F}(\Omega)$ meaning there is no compact extension space.

One potential way to address this issue would be to only consider bounded non-linearities such as the sigmoid or arctan in the second network as a way to control the wild behavior as cloud size gets large. Even so, it is not immediately clear what the natural underlying compact Hausdorff space for such a model would be and it may very well depend on the activation functions. Alternative tools that do not depend on compactness may be needed.

## 4 Limitations of PointNets and Normalized-DeepSets

Unlike the classical UAT, we cannot expect to be able to approximate all continuous functions. This is because $\mathcal{U}(M) \subsetneq \mathcal{C}(M)$ for non-compact metric spaces $M$, and both $\mathcal{F}_H(\Omega)$ and $\mathcal{F}_W(\Omega)$ are non-compact when $(\Omega, d)$ is an infinite compact metric space e.g. the Euclidean unit ball. Since the Point-Cloud-UAT says we can only approximate the uniformly continuous functions, this will immediately impose some surprising limitations on the approximation power of PointNet and normalized-DeepSets.

The following is a counter-intuitive limitation that occurs when considering arbitrarily large point clouds. Define the point cloud diameter and center-of-mass functions by $\mathrm{Diam}(A) = \max_{\boldsymbol{x},\boldsymbol{y} \in A} d(\boldsymbol{x}, \boldsymbol{y})$ and $\mathrm{Cent}(A) = \frac{1}{|A|} \sum_{\boldsymbol{x} \in A} \boldsymbol{x}$.

**Theorem 4.1.** *Suppose $(\Omega, d)$ has no isolated points. Then $f : \mathcal{F}(\Omega) \to \mathbb{R}$ is continuous with respect to both $d_H$ and $d_W$ iff it is constant. Thus, for $\Omega = [0,1]^n$, there is no non-constant function on $\mathcal{F}(\Omega)$ which can be uniformly approximated by both PointNets and normalized-DeepSets. In particular, PointNets can uniformly approximate $\mathrm{Diam}$ but not $\mathrm{Cent}$ and normalized-DeepSets can uniformly approximate $\mathrm{Cent}$ but not $\mathrm{Diam}$.*

At a high level, this dramatic disagreement in approximation power over $\mathcal{F}(\Omega)$ stems from how $d_H$ and $d_W$ handle infinite cardinality limits. Both metrics allow for sequences $A_n \in \mathcal{F}(\Omega)$ with $\lim_{n \to \infty} |A_n| = \infty$ which nonetheless converge to some finite $A \in \mathcal{F}(\Omega)$. However, they in general do not agree on what the the limiting point cloud $A$ should be. As a result, which functions count as continuous "at infinity," and hence on all of $\mathcal{F}(\Omega)$, are radically different.

While it is interesting to know that these neural networks describe dramatically different kinds of functions in the unbounded cardinality setting, in practice there is always a bound due to computational limitations. Nevertheless, the above result sheds some light on the differences between PointNet and normalized-DeepSets in the limit of large cloud size. The next results show that under mild assumptions, and even when we bound point cloud cardinality, PointNet still cannot uniformly approximate many interesting yet simple functions.

**Theorem 4.2.** *Suppose $(\Omega, d)$ is has no isolated points, $f \in \mathcal{C}(\Omega, \mathbb{R}^N)$, and $F \in \mathcal{C}(\mathcal{F}_H^{\leq k}(\Omega), \mathbb{R}^N)$. Then for every $p, q \in \Omega$ and $\tau \in (0, 1]$ there exists a $k$-point set $A$ such that $p, q \in A \subseteq \Omega$ and*

$$\|F(A) - \mathrm{ave}_f(A)\| \geq (1 - \tau) \left( \frac{k-2}{2k} \right) \|f(p) - f(q)\|.$$

*In terms of supremum norm, this simplifies to*

$$\|F - \mathrm{ave}_f\|_{\mathcal{F}^k(\Omega)} \geq \frac{k-2}{2k} \mathrm{Diam}(f(\Omega)).$$

*In particular, this error bound inequality applies to PointNets. Thus for $k \geq 3$ and $f : [0,1]^n \to \mathbb{R}^N$ continuous, $\mathrm{ave}_f$ is uniformly approximable by PointNets on $\mathcal{F}^k([0,1]^n)$ iff $f$ is constant.*[4]

This theorem considerably sharpens prior results on what cannot be represented by PointNet. In the proof of Theorem B.1 of [26], the fixed cardinality case for $\Omega = [0,1]$ is considered and it is shown that $\mathrm{sum}_{\mathrm{id}}(A) = \sum_{a \in A} a$ cannot be *exactly* represented as $\rho \circ \max_f$ when the output dimension of $f$ is too small. With Theorem 4.2, we can now say (without any additional assumptions on the architecture details) that PointNets cannot even *approximate* $\mathrm{sum}_f$ and $\mathrm{ave}_f$ if $f$ is non-constant. Moreover, Theorem 4.2 can quantify how badly attempts at approximation must fail.

Notably, the function class which is being missed by PointNet is very broad and contains some important functions. For example in 1D, if $f(x) = x^m$ then $\mathrm{ave}_f$ computes the $m$-th statistical

---

[4]Note that we can obtain a version of this result for $\mathrm{sum}_f$ by dividing both sides by $k$.

moment. This suggest that if the learning task requires making use of statistical properties of the point cloud that PointNet may struggle or even fail (see Fig. 2 and Sec. 5). By the fixed cardinality universality result of [31], this implies that normalized-DeepSets and standard DeepSets are strictly more expressive function classes than PointNet in the fixed cardinality setting.

For some intuition as to why the result works, let's sketch why PointNet cannot uniformly approximate Cent. Let $p \neq q$ be two points in a $k$-point set. Continuously moving and merging the other $k - 2$ points entirely to $p$ or entirely to $q$ results in the same 2-element set $\{p, q\}$ in a $d_H$-continuous way. By $d_H$-continuity, this means PointNet must output similar predictions for the center-of-mass as we approach this 2-element set along each of these two paths. However, this will lead to conflicting estimates for the center of mass along the way. Quantifying this discrepancy leads to Theorem 4.2.

Upon an initial reading of Theorem 1 of [18] it is easy to overstep and conclude that *every* continuous permutation-invariant function is uniformly approximable by PointNets. However, this is not the case. Moreover, in light of Theorem 4.2 we can see that this fails to be true for a large class of functions.

## 5   Experiments

In all the experiments, cloud cardinality is fixed so there is no theoretical difference between DeepSets and normalized-DeepSets and so we will use the terms interchangeably.

**Testing the Error Lower Bound.** The proof of Theorem 4.2 not only establishes the error bound but also suggest an algorithmic approach to finding point clouds that exhibit the failure of uniform approximation. This let's us produce difficult examples for the centor-of-mass problem for PointNet (even if the weights change). When $\Omega$ is additionally convex – e.g. the unit disk $D$ in $\mathbb{R}^2$ – it becomes fairly easy to construct many examples of $A_p^\delta$ and $A_q^\delta$ explicitly for a given PointNet model, allowing us to empirically verify the uniform-norm error lower bound. In the following experiment, we train a simple PointNet architecture to learn the center-of-mass for 10-element point clouds in $D$. We train on a synthetic dataset of 1 million point

Figure 1: Generated problematic examples (blue) for a PointNet trained to compute the center-of-mass. The method always produces errors at least as large as the theoretical guarantee (orange).

clouds (each element uniformly sampled from $D$) labeled with their center-of-mass. The PointNet architecture has 500K trainable parameters. The network has the form $F(A) = \boldsymbol{\rho}(\max_{\boldsymbol{a} \in A} \boldsymbol{\varphi}(a))$ where $\boldsymbol{\varphi}$ has 2-D input layer, 500-D hidden layer, and 500-D linear output layer, and $\boldsymbol{\rho}$ has 500-D input layer, 500-D hidden layer and 2-D linear output layer (in accordance with the Point-Cloud-UAT). The hidden layer activation functions of $\boldsymbol{\varphi}$ and $\boldsymbol{\rho}$ are ReLU. Since it is not possible to train with respect to the uniform-norm, we opt to train with the traditional $L^2$ loss.

To form our problematic examples, we pick a nonzero $\tau = 0.01$ and two distinct points $p, q \in D$ at random. We set $\delta = \tau \|p - q\| / 4$. We then sample $D$ another 8 times and then linearly pull those 8 points towards $p$ and towards $q$ to within $\delta$. This $\{p, q\}$ adjoined with the 8 points pulled towards $p$ and $q$ form $A_p^\delta$ and $A_q^\delta$ (resp). So that the criteria in the proof of Theorem 4.2 is satisfied, we continue pulling the 8 points closer until $F(A_p^\delta)$ and $F(A_q^\delta)$ are within $\delta$ of $F(\{p, q\})$. We are theoretically ensured one of these two will have error larger than our bound. In Fig. 1 we plot the the produced error vs $\|p - q\|$ for the $p, q \in D$ that were used in $A_p^\delta$ and $A_q^\delta$. As predicted, all the errors for the discovered problematic examples lie above the line representing the uniform-norm error lower bound.

**Lipschitz Constants and Inductive Bias.** Theorem 3.7 showed us that PointNet and normalized-DeepSets are Lipschitz with respect to $d_H$ and $d_W$ respectively. Intuitively, if $d_H(A, B)$ is small this means that $A$ and $B$ have similar *shapes*, and if $d_W(A, B)$ is small then this means $A$ and $B$ has similar *distribution* of points (in the statistical sense). In this experiment we create two synthetic binary classification datasets: one where the two classes have similar shape but differing distributions

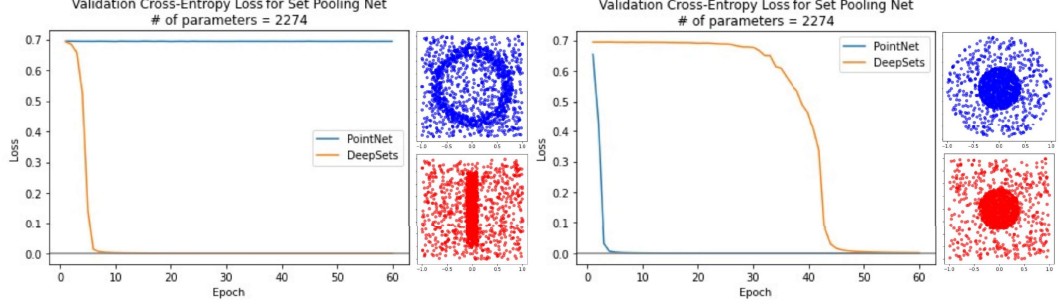

(a) Similar shape, different distributions        (b) Similar distributions, different shapes

Figure 2: Learning curves for **RingLine** (left) and **DiscSquare** (right) binary classification datasets.

(Fig. 2a), and one where the two classes have similar distribution but differeing shapes (Fig. 2b). These datasets are designed to produce advantages and disadvantages for PointNet and DeepSets.

Each dataset contains 2000 examples per class, and the per-example cardinality is kept fixed to 1500 points per cloud for both datasets. The first dataset, **RingLine**, is comprised of 750 samples from a unit square + 750 samples from a small ring or small vertical line (see right panel of Fig. 2a). The class labels are determined by whether the point cloud contains a ring or a line. Here the main difference is the internal *distribution* of points because the point clouds both have square *shape*. The second dataset, **DiscSquare**, is comprised of 1000 samples from a small internal disc + 500 samples from an ambient large disc or large square (see right panel of Fig. 2b). The class labels are determined by whether the ambient shape was a disc or square. Here the most salient difference is the cloud *shape* because the interal *distributions* are similar (concentrated in the center, sparse elsewhere).

In each experiment we train PointNet and DeepSets to correctly classify the two classes on an 80-20 train-test split. The set pooling networks involved have identical architecture except for the choice of pooling (max vs average). In each experiment, we train both for 60 epochs with cross-entropy loss via SGD with learning rate 0.1 and 32 point clouds per batch.

When training on **RingLine**, DeepSets rapidly learns how to perfectly classify the the dataset, but PointNet never progresses beyond random guessing (Fig. 2a). Because PointNet is fundamentally incapable of approximating $\mathrm{ave}_f$ (Theorem 4.2), it struggles to compute anything which could help it distinguish two squares with different internal distributions. Conversely, PointNet rapidly learns to perfectly classify **DiscSquare**, but DeepSets takes about 10x as long to reach the same performance (Fig. 2b). Notably, in this case DeepSets does eventually learn how to solve the dataset, but languishes for a long while. We suspect this has to do with the fact that DeepSets has more expressive power than PointNet. This let DeepSets eventually learn the right thing to do, but it also means it had to waste many epochs trying out useless possibilities. On the **DiscSquare** dataset, PointNet's inability to see beyond shape became a powerful inductive bias.

# 6 Conclusion

The failure of the perceptron to learn XOR was a blow that raised the spectre of limited representation power until UATs arose to ease concerns. In this paper we studied analogous impossibility and universality questions for three kinds of set pooling networks. By carefully choosing the underlying topologies and metrics we showed how the choice of pooling function can have a dramatic impact on the expressivity of these networks. In future works it would be valuable to further unify, systematize, and generalize this topological approach to handle all pooling functions and perhaps even other architectures such as transformers.

# 7 Acknowledgements

The first author would like to acknowledge partial support from grant DOE ASCR PhILMS DE-SC0019246, as well as support from NASA SCAN during their internships at NASA Glenn Research Center. Additional thanks to Arturo Deza, Garo Sarajian, and Steve Trettel for invaluable discussions.

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
