# Appendix

Here we prove all the claims for which there was insufficient room for the proof in the main body of the paper. Please refer to the main paper for definitions which were previously described or to the references for any topics which are standard.

## A    Uniform Closure of Extensions Lemma

We will make use of the following simple lemma about continuous extensions in the proof of the Point-Cloud-UAT (Theorem 3.4).

**Lemma A.1.** *Let $\mathcal{N} \subseteq \mathcal{B}(D)$ be a family of bounded functions and $D$ a dense subset of a compact Hausdorff space $X$. Suppose $\mathcal{N}$ has a continuous extension to $X$ denoted by $\mathcal{N}' \subseteq \mathcal{C}(X)$ which is dense. Then the uniform closure of $\mathcal{N}$ in $\mathcal{B}(D)$ is $r(\mathcal{C}(X))$ where $r : \mathcal{C}(X) \to \mathcal{C}_b(D)$ is the domain restriction map. If $X$ can be metrized by a metric d, then $r(\mathcal{C}(X)) = \mathcal{U}(D)$.*

*Proof.* First we show that the $r$ is a linear isometry. Since $X$ is compact for $f \in \mathcal{C}(X)$ there is a $p \in X$ so that $|f(p)| = \|f\|_X$. By density of $D$ there is a sequence $p_n \in D$ that limits to $p$. So

$$\|f\|_X = |f(p)| = \lim_{n \to \infty} |f(p_n)| \leq \sup_{x \in D} |f(x)| = \|r(f)\|_D \leq \sup_{x \in X} |f(x)| = \|f\|_X$$

Next, since $\mathcal{C}(X)$ is complete, so is its isometric image $r(\mathcal{C}(X))$ and because $\mathcal{C}_b(X)$ is complete that means $r(\mathcal{C}(X))$ is closed. Thus,

$$r(\mathcal{C}(X)) = r(\overline{\mathcal{N}'}) \subseteq \overline{r(\mathcal{N}')} \subseteq \overline{r(\mathcal{C}(X))} = r(\mathcal{C}(X)),$$

where the first subset results from continuity. Thus $\overline{\mathcal{N}} = \overline{r(\mathcal{N}')} = r(\mathcal{C}(X))$.

Finally, $\mathcal{U}(D) \subseteq r(\mathcal{C}(X))$ because every uniformly continuous function on a dense set extends to uniformly continuous function on the whole space. The reverse inclusion follows because restriction preserves uniform continuity. $\square$

Letting $D = \mathcal{F}(\Omega)$, this lemma suggest the following plan of attack for proving the Point-Cloud-UAT: find a compact metric space $(X, d)$ in which we can realize $\mathcal{F}(\Omega)$ as a dense subset and hope that our class of neural networks $\mathcal{N}$ continuously extends to a dense subset of $\mathcal{C}(X)$. If we can do that, then we know the uniform closure of our class of neural networks are precisely the uniformly continuous functions on $\mathcal{F}(\Omega)$ with respect to the metric inherited from $X$. We accomplish this for PointNet and normalized-DeepSets on $\mathcal{F}(\Omega)$ by utilizing $(\mathcal{K}(\Omega), d_H)$ and $(\mathcal{P}(\Omega), d_W)$.

## B    Continuous Extension Lemma

**Lemma 2.1.** *Let $(\Omega, d)$ be compact, $f \in \mathcal{C}(\Omega)$. Then $\mathrm{Max}_f \in \mathcal{C}(\mathcal{K}(\Omega))$ and $\mathrm{Ave}_f \in \mathcal{C}(\mathcal{P}(\Omega))$ and $\mathrm{Max}_f \circ i_{\mathcal{K}} = \max_f$ and $\mathrm{Ave}_f \circ i_{\mathcal{P}} = \mathrm{ave}_f$. As a consequence, PointNet and normalized-DeepSets are uniformly continuous on $\mathcal{F}_H(\Omega)$ and $\mathcal{F}_W(\Omega)$ respectively.*

*Proof.* First we show that $\mathrm{Max}_f$ is $d_H$-continuous. Let $\epsilon > 0$. Since $\Omega$ is compact, $f$ is uniformly continuous and so there is a $\delta > 0$ so that $|f(x) - f(y)| < \epsilon/2$ whenever $d(x, y) < 2\delta$. Now let $A, B \in \mathcal{K}(\Omega)$ and suppose $d_H(A, B) < \delta$. By definition this means $A \subseteq B_\delta$ and $B \subseteq A_\delta$. By the triangle inequality we have

$$|\mathrm{Max}_f(A) - \mathrm{Max}_f(B)| \leq |\mathrm{Max}_f(A) - \mathrm{Max}_f(A_\delta)| + |\mathrm{Max}_f(A_\delta) - \mathrm{Max}_f(B)|.$$

Since $A_\delta$ is compact there is a $p \in A_\delta$ so that $\mathrm{Max}_f(A_\delta) = f(p)$. Observe that if $q \in K \subseteq A_\delta$ with $d(p, q) < 2\delta$, then $|f(p) - f(q)| < \epsilon/2$ and $f(p) = \mathrm{Max}_f(A_\delta) \geq \mathrm{Max}_f(K) \geq f(q)$. This implies $|\mathrm{Max}_f(A_\delta) - \mathrm{Max}_f(K)| < \epsilon/2$ whenever we can find such a $q \in K$. For $K = A$, note that since $p \in A_\delta$ there is an $a \in A \subseteq A_\delta$ such that $d(p, a) < \delta < 2\delta$, so $|\mathrm{Max}_f(A_\delta) - \mathrm{Max}_f(A)| < \epsilon/2$. For $K = B$, note that $B$ is compact so there is a $b \in B \subseteq A_\delta$ closest to $p$, and so

$$d(p, b) \leq d_H(A_\delta, B) \leq d_H(A_\delta, A) + d_H(A, B) < 2\delta,$$

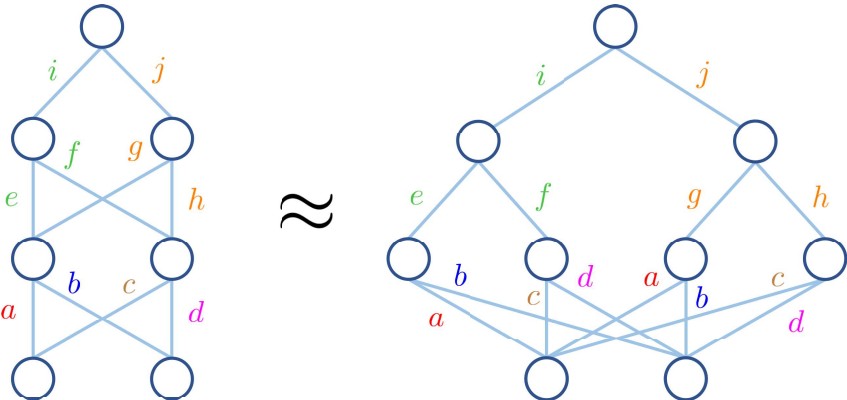

Figure 3: Two equivalent networks with different architectures that are equal as functions.

which means $|\text{Max}_f(A_\delta) - \text{Max}_f(B)| < \epsilon/2$. Thus, $|\text{Max}_f(A) - \text{Max}_f(B)| < \epsilon$ as desired.

To see why $\text{Ave}_f$ is $d_W$-continuous recall that the topology of $d_W$ is the same as the weak-* topology for measures and so the map $\mu \mapsto \int f \, d\mu$ is by definition continuous whenever $f \in \mathcal{C}(\Omega)$. By definitions and linearity we have,

$$\text{Ave}_f(i_\mathcal{P}(A)) = \int f \, d\left(\frac{1}{|A|}\sum_{a \in A}\delta_a\right) = \frac{1}{|A|}\sum_{a \in A}\int f \, d\delta_a = \frac{1}{|A|}\sum_{a \in A}f(a) = \text{ave}_f(A).$$

Next, note that $\text{Max}_f \circ i_\mathcal{K} = \max_f$ is trivially true.

Lastly, by composition of continuous functions it follows that PointNet and normalized-DeepSets continuously extend to $\mathcal{K}(\Omega)$ and $\mathcal{P}(\Omega)$. Since $(\Omega, d)$ compact implies both $\mathcal{K}(\Omega)$ and $\mathcal{P}(\Omega)$ are compact, we can deduce that PointNet and normalized-DeepSets are uniformly continuous on $\mathcal{F}_H(\Omega)$ and $\mathcal{F}_W(\Omega)$. $\qquad\square$

## C  Explanation of Neural Network Notation

The neural network notation of Sec. 2.4 deliberately trades-off detailed information about network architectures (e.g. number of weights/neurons and their organization) in exchange for making the mathematical properties of the function class they represent more transparent to help with proofs. Architecture details can be distracting and cumbersome for our purposes as different neural networks with different architectures but equal number of layers can represent exactly the same function as in Fig. 3.

To clarify the notation, let us revisit $\mathcal{N}^\sigma$ which is defined in Sec. 2.4. If we let Aff denote the set of all affine functionals (i.e. $\mathbb{R}$-valued functions of the form $f(\boldsymbol{x}) = \boldsymbol{w} \cdot \boldsymbol{x} + b$) then it is easily verifiable that $\mathcal{N}^\sigma = \text{span}(\sigma \circ \text{Aff})$. In the definition of the LHS in we gave an explicit formula that elements of $\mathcal{N}^\sigma$ must satisfy (Sec. 2.4). On the other hand, the approach taken by the RHS captures the same functions with less notation and emphasizes that this family of functions is closed under linear combinations.

To go deeper with this notation, we can simply compose already constructed shallower networks with an activation function $\tau$ and then take all possible linear combination of such things. Using the notation in Sec. 2.4 this is expressed by $\mathcal{N}^{\sigma,\tau} = \text{span}\left(\tau \circ \mathcal{N}^\sigma\right)$. To get another layer, we write $\mathcal{N}^{\sigma,\tau,\rho} = \text{span}\left(\rho \circ \mathcal{N}^{\sigma,\tau}\right)$. To determine the number of hidden layers involved we can simply count the number of activation functions. However, it is not possible to determine the number of weights as members of these families have arbitrarily wide layers.

In Theorem 3.4 we show that $\mathcal{N}_{\max}^{\sigma|\tau}$ and $\mathcal{N}^{\sigma|\tau_{\text{ave}}}$ are universal for $\mathcal{U}(\mathcal{F}_H(\Omega))$ and $\mathcal{U}(\mathcal{F}_W(\Omega))$ respectively. To unpack this notation, let $\max_{\mathcal{N}^\sigma} = \{\max_f \mid f \in \mathcal{N}^\sigma\}$ and $\text{ave}_{\mathcal{N}^\sigma} = \{\max_f \mid f \in \mathcal{N}^\sigma\}$.

Note that both $\max_{\mathcal{N}^\sigma}$ and $\mathrm{ave}_{\mathcal{N}^\sigma}$ both have one hidden-layer and a max-pooling output-layer.[5] With this we can see that,

$$\mathcal{N}^{\sigma|\tau}_{\max} = \mathrm{span}\left(\tau \circ \mathcal{N}^\sigma_{\max}\right), \qquad\qquad \mathcal{N}^{\sigma|\tau}_{\mathrm{ave}} = \mathrm{span}\left(\tau \circ \mathcal{N}^\sigma_{\mathrm{ave}}\right),$$
$$= \mathrm{span}\left(\tau \circ \mathrm{span}\left(\max_{\mathcal{N}^\sigma}\right)\right), \qquad\qquad = \mathrm{span}\left(\tau \circ \mathrm{span}\left(\mathrm{ave}_{\mathcal{N}^\sigma}\right)\right),$$
$$= \mathrm{span}\left(\tau \circ \left(\mathrm{ave}_{\mathcal{N}^\sigma}\right)\right).$$

Note that the expression for normalized-DeepSets (average-pooling) simplifies but the expression for PointNet does not. This is because $\mathrm{ave}_{\mathcal{N}^\sigma}$ is closed under linear combinations (as mentioned in Sec. 2.4). By examining these expressions, we can see that: (1) $\max_{\mathcal{N}^\sigma}$ accounts for an input-layer, a hidden-layer, and a max-pooling layer, (2) the ensuing $\mathrm{span}(\cdot)$ creates another layer, (3) the application of $\tau$ applies an activation to this new layer but keeps layer count the same, and (4) the last $\mathrm{span}(\cdot)$ creates the $\mathbb{R}$-valued output layer. This yields a total of three hidden-layers. A similar story unfolds for $\mathcal{N}^{\sigma|\tau}_{\mathrm{ave}}$ except that the internal simplification allows us to skip the linear combination in step (2) and thus we are able to get away with just two hidden-layers.

## D  Topological UAT

**Theorem 3.1** (Topological-UAT). *Let $X$ be a compact Hausdorff space and $\sigma \in \mathfrak{U}(\mathbb{R})$. If $S \subseteq \mathcal{C}(X)$ separates points and contains a nonzero constant, then $\mathrm{span}(\sigma \circ \mathrm{span}\, S)$ is dense in $\mathcal{C}(X)$. Additionally, if $S$ also happens to be a linear subspace, then $\mathrm{span}(\sigma \circ S)$ is dense in $\mathcal{C}(X)$.*

*Proof.* Let $S$ and $\sigma$ satisfy the above and let $V = \mathrm{span}\, S$. Let $\mathrm{Alg}(V)$ denote the algebra generated by $V$, i.e. all possible finite products, sums and scalar multiples of the elements of $V$. Then $\mathrm{Alg}(V)$ is unital subalgebra of $\mathcal{C}(X)$ that seperates points. By the Stone-Weierstrass theorem $\mathrm{Alg}(V)$ is dense in $\mathcal{C}(X)$. Now let $F \in \mathcal{C}(X)$ and $\epsilon > 0$ be arbitrary. By density there is a $G \in \mathrm{Alg}(V)$ such that $|F(a) - G(a)| < \epsilon/2$ for all $a \in X$. Since $G \in \mathrm{Alg}(V)$ there is an $N$-variable polynomial $p$ and $\boldsymbol{s} = (s_1, \ldots, s_N)$ where $s_i \in S$, so that $G = p \circ \boldsymbol{s}$. Since all $s_i \in \mathcal{C}(X)$ and $X$ is compact, the image $\boldsymbol{s}(X) \subseteq \mathbb{R}^N$ is compact. By the classical UAT [14], there exists an $\eta \in \mathcal{N}^\sigma$ such that $|p(\boldsymbol{x}) - \eta(\boldsymbol{x})| < \epsilon/2$ for all $\boldsymbol{x} \in \mathbb{R}^N$. Thus,

$$|F(a) - (\eta \circ \boldsymbol{s})(a)| \leq |F(a) - p(\boldsymbol{s}(a))| + |p(\boldsymbol{s}(a)) - \eta(\boldsymbol{s}(a))| < \epsilon/2 + \epsilon/2 = \epsilon$$

for every $a \in X$. Finally note that $\eta(\boldsymbol{s}(a)) = \sum_{i=1}^{m} a_i \sigma(\boldsymbol{w}_i \cdot \boldsymbol{s}(a) + b_i)$ for some $a_i, b_i \in \mathbb{R}$ and $\boldsymbol{w}_i \in \mathbb{R}^N$. Since $S$ constains a nonzero constant, $\mathrm{span}\, S$ contains every constant and so $\boldsymbol{w}_i \cdot \boldsymbol{s} + b_i \in \mathrm{span}\, S$. Thus $\eta \circ \boldsymbol{s} \in \mathrm{span}(\sigma \circ \mathrm{span}\, S)$ as desired.

Lastly, if $S$ is also linear subspace, then $S = \mathrm{span}\, S$ and so $\mathrm{span}(\sigma \circ S)$ is dense in $\mathcal{C}(X)$. $\qquad\square$

## E  Separating Points with $\mathrm{Max}_f$ and $\mathrm{Ave}_f$

**Lemma 3.2** (Separation Lemma). *Let $\Omega \subseteq \mathbb{R}^N$ be compact and $\sigma \in \mathfrak{U}(\mathbb{R})$. Then the set of functions $S_{\mathrm{Max}} = \{\mathrm{Max}_f \mid f \in \mathcal{N}^\sigma\}$ and $S_{\mathrm{Ave}} = \{\mathrm{Ave}_f \mid f \in \mathcal{N}^\sigma\}$ separate points and contain constants.*

*Proof.* Let $d$ denote the Euclidean distance. First note that by choosing weights correctly, we can find a constant function $h = \sigma(c) \in \mathcal{N}^\sigma$ for some $c \in \mathbb{R}$. Since $\sigma$ is not a polynomial, there is a choice of $c$ for which $\sigma(c) \neq 0$. This means $\mathrm{Max}_h \in S_{\mathrm{Max}}$ and $\mathrm{Ave}_h \in S_{\mathrm{Ave}}$ are both constant. Now we just need to show that $S_{\mathrm{Max}}$ and $S_{\mathrm{Ave}}$ separate points.

($S_{\mathrm{Max}}$ separates points): Let $A, B \in \mathcal{K}(\Omega)$ with $A \neq B$. Without loss of generality, $A \setminus B \neq \varnothing$ so choose $a \in A \setminus B$. Let $f(x) = \min\{1, d(x, B)/d(a, B)\}$ and note that $f(a) = 1$, $f(B) = \{0\}$ and $f(\Omega) = [0, 1]$. By the classical UAT [14] $\mathcal{N}^\sigma$ is dense in $\mathcal{C}(\Omega)$, so there is a $g \in \mathcal{N}^\sigma$ so that $|f(x) - g(x)| < 1/2$ for all $x \in \Omega$. Note $\mathrm{Max}_g \in S_{\mathrm{Max}}$ and that $\mathrm{Max}_g(A) > 1/2$ and $\mathrm{Max}_g(B) < 1/2$. Since $A$ and $B$ were arbitrary, this shows $S_{\mathrm{Max}}$ separates point in $\mathcal{K}(X)$.

($S_{\mathrm{Ave}}$ separates points): Given $\mu_1, \mu_2 \in \mathcal{P}(\Omega)$ with $\mu_1 \neq \mu_2$, by the Hahn-Banach separation theorem there exists a weak-* continuous linear functional $L : M(\Omega) \to \mathbb{R}$ that separates them. Let

---

[5]Depending on convention, one can count this as "input layer + two hidden layers + max-pooling" or "input layer + one hidden-layer + max-pooling output layer." We opt for the latter, but the interpretation is ultimately up to the reader.

$\delta = |L(\mu_1) - L(\mu_2)|$. The topological dual of $M(\Omega)$ with the weak-* topology is equivalent to $\mathcal{C}(\Omega)$ and so there is an $f \in \mathcal{C}(\Omega)$ so that $L(\eta) = \int f \, d\eta$ for all $\eta \in M(\Omega)$. Since $\mathcal{N}^\sigma$ is dense in $\mathcal{C}(\Omega)$ there is a $g \in \mathcal{N}^\sigma$ so the that $|f(x) - g(x)| < \delta/2$ for all $x \in \Omega$. Define $J(\eta) = \int g \, d\eta$. Then for all $\eta \in \mathcal{P}(\Omega)$ we have $|L(\eta) - J(\eta)| \le \int |f - g| \, d\eta < \frac{\delta}{2} \int d\eta = \delta/2$. Applying the triangle inequality we obtain

$$\delta \le \underbrace{|L(\mu_1) - J(\mu_1)|}_{<\delta/2} + |J(\mu_1) - J(\mu_2)| + \underbrace{|J(\mu_2) - L(\mu_2)|}_{<\delta/2}.$$

Thus $0 < |J(\mu_1) - J(\mu_2)|$ and so $J = \text{Ave}_g \in S_{\text{Ave}}$ separates $\mu_1$ and $\mu_2$. Since $\mu_1$ and $\mu_2$ were arbitrary, it follows that $S_{\text{Ave}}$ seperates points in $\mathcal{P}(\Omega)$. $\qquad\square$

# F  Lipschitz Continuity of PointNet and Normalized-DeepSets

**Lemma 3.6.** *Suppose $(\Omega, d)$ is compact and $f : \Omega \to \mathbb{R}$ is Lipschitz continuous with Lipschitz constant $L$. Then $\text{Max}_f \in \mathcal{K}(\Omega)$ and $\text{Ave}_f \in \mathcal{P}(\Omega)$ have Lipschitz constant $2L$ and $L$.*

*Proof.* Suppose $|f(x) - f(y)| \le L \, d(x, y)$ and let $A, B \in \mathcal{K}(\Omega)$ and $\mu, \nu \in \mathcal{P}(\Omega)$ be arbitrary. First we will prove the Lipschitz bound for $\text{Max}_f$ and then for $\text{Ave}_f$.

Let $C = A \cup B$. Since $C$ is compact, there exists a $c \in C$ such that $\text{Max}_f(C) = f(c)$. Since $A \subseteq C$, then by definition of $\text{Max}_f$ we have $f(c) = \text{Max}_f(C) \ge \text{Max}_f(A) \ge f(a)$ for any $a \in A$. Thus, $\text{Max}_f(C) - \text{Max}_f(A) \le f(c) - f(a)$ and so

$$|\text{Max}_f(C) - \text{Max}(A)| \le |f(c) - f(a)| \le L \, d(c, a).$$

Now let $a^* \in A$ be the point in $A$ closest to $c$. Then since $A \subseteq C$,

$$|\text{Max}_f(C) - \text{Max}_f(A)| \le L \, d(c, a^*) \le L \, d_H(C, A).$$

Since $B \subseteq C$, the same argument yields $|\text{Max}_f(C) - \text{Max}_f(B)| \le L \, d_H(C, B)$. Thus, by the triangle in equality and $d_H(A, A \cup B), d_H(B, A \cup B) \le d_H(A, B)$ (follows from [2] Theorem 1.12.15) we have

$$\begin{aligned}
|\text{Max}_f(A) - \text{Max}_f(B)| &\le |\text{Max}_f(A) - \text{Max}_f(C)| + |\text{Max}_f(C) - \text{Max}_f(B)| \\
&\le L \, d_H(A, A \cup B) + L \, d_H(A \cup B, B) \\
&\le 2L \, d_H(A, B).
\end{aligned}$$

Thus $\text{Max}_f$ has Lipschitz constant $2L$ as desired.

Next we consider $\text{Ave}_f$. Note that $\widetilde{f} = \frac{1}{L} f$ is 1-Lipschitz so by the Kantorovich-Rubenstein duality we have the desired result.

$$|\text{Ave}_f(\mu) - \text{Ave}_f(\nu)| = L \left| \int \widetilde{f} \, d\mu - \int \widetilde{f} \, d\nu \right| \le L \left| \sup_{\|g\|_{\text{Lip}} \le 1} \int g \, d\mu - \int g \, d\nu \right| = L \, d_W(\mu, \nu).$$

$\qquad\square$

The Kantorovich-Rubenstein duality plays a big role in making the Lipschitz behaviour of normalized-DeepSets apparent. It is here that we can see the importance of using the 1-Wasserstein distance instead of other $p$-Wasserstein distance.

**Theorem 3.7.** *Suppose $\Omega \subseteq \mathbb{R}^N$ is compact. Then every PointNet and normalized-DeepSets network with Lipschitz activation functions is Lipschitz on $\mathcal{F}_H(\Omega)$ and $\mathcal{F}_W(\Omega)$ respectively.*

*Proof.* Recall that a PointNet network is of the form $\psi = \rho \circ \max_f$ for neural networks $f : \Omega \to \mathbb{R}^k$ and $\rho : \mathbb{R}^k \to \mathbb{R}^N$. Since both $f$ and $\rho$ are an alternating composition of affine transforms and nonlinear transforms, and all of these are Lipschitz, it follows that both $f$ and $\rho$ are Lipschitz. By Lemma 3.6, $f$ being Lipschitz implies $\max_f$ is Lipschitz. Thus $\psi$ is Lipschitz since it is a composition of Lipschitz functions. The analogous argument applies for the case of normalized-DeepSets. $\qquad\square$

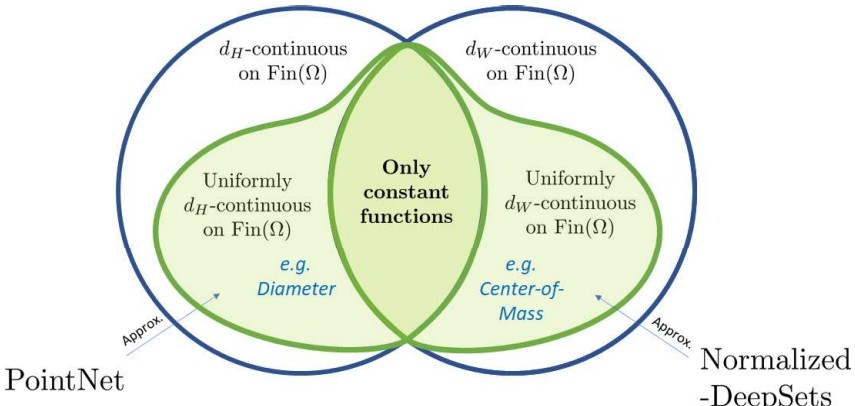

Figure 4: A Venn diagram summarizing Theorem 3.4 and Theorem 4.1 by showing how PointNet and normalized-DeepSets relate to he different function classes they can and can't uniformly approximate.

## G    Unbounded Cardinality Limitation Theorem

**Theorem 4.1.** *Suppose* $(\Omega, d)$ *has no isolated points. Then* $f : \mathcal{F}(\Omega) \to \mathbb{R}$ *is continuous with respect to both* $d_H$ *and* $d_W$ *iff it is constant. Thus, for* $\Omega = [0, 1]^n$, *there is no non-constant function on* $\mathcal{F}(\Omega)$ *which can be uniformly approximated by both PointNets and normalized-DeepSets. In particular, PointNets can uniformly approximate* Diam *but not* Cent *and normalized-DeepSets can uniformly approximate* Cent *but not* Diam.

*Proof.* Assume $f : \mathcal{F}(\Omega) \to \mathbb{R}$ is both $d_H$-continuous and $d_W$-continuous. Let $A \in \mathcal{F}(\Omega)$ and let $p \in A$. For each $n = 1, 2, \ldots$, choose $A'_n \in \mathcal{F}(\Omega)$ to be an $n$-point set contained within the $1/n$-ball around $p$. We can do this because $\Omega$ has no isolated points. Now let $A_n = A'_n \cup (A \setminus \{p\})$.

Observe that $A_n \overset{d_H}{\to} A$ and $A_n \overset{d_W}{\to} \{p\}$. Thus by continuity,

$$f(A) = f\left(\overset{d_H}{\underset{n\to\infty}{\lim}} A_n\right) = f\left(\overset{d_W}{\underset{n\to\infty}{\lim}} A_n\right) = f(\{p\}).$$

Note that $A$ was arbitrary so $f$ must always assign a set and any of its singleton subsets the same value. Now let $B, C \in \mathcal{F}(\Omega)$ and let $q \in B$ and $r \in C$ then

$$f(B) = f(\{q\}) = f(\{q, r\}) = f(\{r\}) = f(C),$$

thus $f$ must be constant. Conversely, constant maps are always continuous.

Now let $\Omega = [0, 1]^n$ and suppose $f : \mathcal{F}(\Omega) \to \mathbb{R}$ can be uniformly approximated by both PointNets and normalized-DeepSets. By Theorem 3.4 we know that PointNets and normalized-DeepSets can only uniformly approximate the uniformly continuous functions on $\mathcal{F}(\Omega)$ with respect to $d_H$ and $d_W$ respectively. Applying the result established above leads us to conclude that that $f$ is uniformly approximable by both iff it is constant.

Finally, it is known that the Diam satisfies $|\mathrm{Diam}(A) - \mathrm{Diam}(B)| \leq 2d_H(A, B)$ and hence is $d_H$-continuous on $\mathcal{K}(\Omega)$ and Cent is $d_W$-continuous on $\mathcal{P}(\Omega)$ because $\mathrm{Ave}_{\pi_i}$ is $d_W$-continuous (here $\pi_i$ is the projection onto the $i$-th component map). This means they are uniformly continuous on $\mathcal{F}_H(\Omega)$ and $\mathcal{F}_W(\Omega)$ respectively and so the result follows from the above and Theorem 3.4. □

## H    PointNet Error Lower Bound

**Theorem 4.2.** *Suppose* $(\Omega, d)$ *is has no isolated points,* $f \in \mathcal{C}(\Omega, \mathbb{R}^N)$, *and* $F \in \mathcal{C}(\mathcal{F}_H^{\leq k}(\Omega), \mathbb{R}^N)$. *Then for every* $p, q \in \Omega$ *and* $\tau \in (0, 1]$ *there exists a* $k$-point set $A$ *such that* $p, q \in A \subseteq \Omega$ *and*

$$\|F(A) - \mathrm{ave}_f(A)\| \geq (1 - \tau)\left(\frac{k-2}{2k}\right)\|f(p) - f(q)\|.$$

*In terms of supremum norm, this simplifies to*

$$\|F - \mathrm{ave}_f\|_{\mathcal{F}^k(\Omega)} \geq \frac{k-2}{2k} \mathrm{Diam}(f(\Omega)).$$

*In particular, this error bound inequality applies to PointNets. Thus for $k \geq 3$ and $f : [0,1]^n \to \mathbb{R}^N$ continuous, $\mathrm{ave}_f$ is uniformly approximable by PointNets on $\mathcal{F}^k([0,1]^n)$ iff $f$ is constant.[6]*

*Proof.* First note that if $k \in \{1, 2\}$, $\tau = 1$, or $f(p) = f(q)$ that the inequality is trivially satisfied. Thus, we can henceforth assume that $k \geq 3$, $\tau \in (0, 1)$, and $f(p) \neq f(q)$.

Let $C_p := \frac{f(p)+f(q)}{k} + (\frac{k-2}{k})f(p)$ and $C_q := \frac{f(p)+f(q)}{k} + (\frac{k-2}{k})f(q)$ and observe that

$$\|C_p - C_q\| = \frac{k-2}{k} \|f(p) - f(q)\| \neq 0.$$

Let $\epsilon := \frac{\tau}{4} \|C_p - C_q\| > 0$. Since $F$ is $d_H$-continuous there exists a $\delta_{\{p,q\}} \in (0, \epsilon]$ such that whenever $d_H(A, \{p, q\}) < \delta_{\{p,q\}}$ we have that $\|F(A) - F(\{p,q\})\| < \epsilon$. Similarly, since $f$ is continuous, there exists $\delta_p, \delta_q \in (0, \epsilon]$ so that $d(a, p) < \delta_p$ implies $\|f(a) - f(p)\| < \epsilon$ and $d(a, q) < \delta_q$ implies $\|f(a) - f(q)\| < \epsilon$. Define $\delta := \min\{\delta_{\{p,q\}}, \delta_p, \delta_q\}$.

Because $\Omega$ has no isolated points, its open balls must have infinitely many points, in particular, $B_\delta(p)$ and $B_\delta(q)$. This ensures the existence of sets $\widetilde{A}_p^\delta \subseteq B_\delta(p) \setminus \{p\}$ and $\widetilde{A}_q^\delta \subseteq B_\delta(q) \setminus \{q\}$ with cardinality $k - 2$. We can then define $k$-point supersets of $\{p, q\}$ given by

$$A_p^\delta := \widetilde{A}_p^\delta \cup \{p, q\} \text{ and } A_p^\delta := \widetilde{A}_p^\delta \cup \{p, q\}.$$

It is easy to see that both $A_p^\delta$ and $A_q^\delta$ are $\delta$-close to $\{p, q\}$ with respect to $d_H$. Thus, by definition of $\delta$ we have

$$\left\|F(A_p^\delta) - F(\{p, q\})\right\| < \epsilon \text{ and } \left\|F(A_q^\delta) - F(\{p, q\})\right\| < \epsilon.$$

Next observe that,

$$\mathrm{ave}_f(A_p^\delta) = \frac{1}{k} \sum_{a \in A_p^\delta} f(a) = \frac{f(p) + f(q)}{k} + \frac{1}{k} \sum_{a \in \widetilde{A}_p^\delta} f(a), \text{ and}$$

$$\mathrm{ave}_f(A_q^\delta) = \frac{1}{k} \sum_{a \in A_q^\delta} f(a) = \frac{f(p) + f(q)}{k} + \frac{1}{k} \sum_{a \in \widetilde{A}_q^\delta} f(a).$$

Since $f(\widetilde{A}_p^\delta) \subseteq B_\epsilon(f(p))$ and $f(\widetilde{A}_q^\delta) \subseteq B_\epsilon(f(q))$, the triangle inequality implies

$$\left\|\mathrm{ave}_f(A_p^\delta) - C_p\right\| = \frac{1}{k} \left\|\sum_{a \in \widetilde{A}_p^\delta} \left(f(a) - f(p)\right)\right\| \leq \frac{1}{k} \sum_{a \in \widetilde{A}_p^\delta} \|f(a) - f(p)\| < \left(\frac{k-2}{k}\right)\epsilon < \epsilon, \text{ and}$$

$$\left\|\mathrm{ave}_f(A_q^\delta) - C_q\right\| = \frac{1}{k} \left\|\sum_{a \in \widetilde{A}_q^\delta} \left(f(a) - f(p)\right)\right\| \leq \frac{1}{k} \sum_{a \in \widetilde{A}_q^\delta} \|f(a) - f(p)\| < \left(\frac{k-2}{k}\right)\epsilon < \epsilon.$$

Now we can consider the triangle in $\mathbb{R}^N$ formed by $C_p, C_q$, and $F(\{p, q\})$. By basic geometry, we know that one of the two side lengths $\|F(\{p, q\}) - C_p\|$ or $\|F(\{p, q\}) - C_q\|$ must be greater than or equal to half the third side length, i.e. greater than $\|C_p - C_q\|/2$. Without loss of generality, let $\|F(\{p, q\}) - C_q\| \geq \|C_p - C_q\|/2$ and then apply the triangle inequality in conjunction with our $\epsilon$ bounds to yield

$$\frac{\|C_p - C_q\|}{2} \leq \left\|F(\{p, q\}) - F(A_q^\delta)\right\| + \left\|F(A_q^\delta) - \mathrm{ave}_f(A_q^\delta)\right\| + \left\|\mathrm{ave}_f(A_q^\delta) - C_q\right\|$$

$$< \left\|F(A_q^\delta) - \mathrm{ave}_f(A_q^\delta)\right\| + 2\epsilon.$$

---

[6]Note that we can obtain a version of this result for $\mathrm{sum}_f$ by dividing both sides by $k$.

By rearranging and substituting $\epsilon = \tau \left\| C_p - C_q \right\| / 4$, we get

$$
\begin{aligned}
\left\| F(A_q^\delta) - \mathrm{ave}_f(A_q^\delta) \right\| &> \frac{\left\| C_p - C_q \right\|}{2} - 2\epsilon \\
&= \frac{\left\| C_p - C_q \right\|}{2} - \tau \frac{\left\| C_p - C_q \right\|}{2} \\
&= \frac{(1-\tau)}{2} \left\| C_p - C_q \right\| \\
&= (1-\tau) \left( \frac{k-2}{2k} \right) \left\| f(p) - f(q) \right\|.
\end{aligned}
$$

Thus $A_q^\delta$ is the promised set which achieves the desired error and proves the first inequality. Taking the limit as $p, q$ spread apart and $\tau \to 0$ we can also get the supremum norm inequality,

$$
\left\| F - \mathrm{ave}_f \right\|_{\mathcal{F}^k(\Omega)} \geq \frac{k-2}{2k} \, \mathrm{Diam}(f(\Omega)).
$$

Note that this inequality applies when $F$ is a PointNet since PointNets are $d_H$-continuous by Lemma 2.1. Now assume $k \geq 3$, $\Omega = [0,1]^n$, and $f : \Omega \to \mathbb{R}^N$ is continuous. If $f$ is not constant, then $\frac{k-2}{2k} \, \mathrm{Diam}(f(\Omega)) > 0$ meaning that no PointNet $F$ can get closer to $\mathrm{ave}_f$ than that. This means $\mathrm{ave}_f$ is not uniformly approximable by PointNets. Conversely, if it were possible to get arbitrarily close to $\mathrm{ave}_f$ via PointNets in the supremum-norm, then the LHS could be made arbitrarily small, implying that $\mathrm{Diam}(f(\Omega)) = 0$. Thus, $f : \Omega \to \mathbb{R}^N$ would have to be constant, as desired. $\qquad \square$