# OpenReview forum: "On the Representation Power of Set Pooling Networks"
_NeurIPS.cc/2021/Conference — NeurIPS 2021 Poster_

### Official Review · Reviewer_FQK3 · 2021-07-15

**Rating:** 7
**Confidence:** 4

**Summary:**

The paper studies the representation power of two prominent set-learning deep architectures: pointnet and deep sts, that differ in their pooling mechanism (max vs sum/mean). While these two architectures are universal when considering constant-size sets, their expressivity when considering sets of varying cardinalities is not yet well understood.
The main contribution of this paper is establishing an understanding of which function spaces can be approximated by each of these architectures when set sizes vary (Both in the unbounded and bounded set size cases).


**Limitations And Societal Impact:**

Discussed.

**Main Review:**

Positive points:
-- the paper addresses an important and real problem: in practice, in many applications, set sizes vary. The paper analyzes the most fundamental set learning architectures in this case.
The authors did a good job writing the paper. I especially like the introduction which discusses all the key issues and presents good motivations.
Negative points:
-- I found it difficult to follow the math from the paper itself without the supp. But this is understandable considering the page limit.

Additional related literature that could be of interest to the authors:
-- Other papers that discuss the universality of set networks. “On Universal Equivariant Set Networks”, “On Learning Sets of Symmetric Elements”.
-- There is an ongoing effort in the graph learning community to understand expressive power with varying graph sizes which seem to be related. “Size-Invariant Graph Representations for Graph Classification Extrapolation” “from local structures to size generalization in graph neural networks”, “Universal invariant and equivariant graph neural networks” “Expressive Power of Invariant and Equivariant Graph Neural Networks” (appendix F).
In particular, there are universality results for high-order graph networks in the variable node set size case. Could be interesting to discuss if this can be leveraged for the set learning case in this paper.

To summarise, this paper studies an important topic in a mathematically rigorous way. I support acceptance,


**Time Spent Reviewing:**

4

---

> ### Author Response · Authors · 2021-08-10
> **Thank you for your review and constructive comments. Below we address some of the raised points.**
>
> **Clarity:** As you surmised, given the page limit it proved challenging to rigorously and accessibly describe the mathematics solely within the confines of the main paper. To address this, we plan to focus on elaborating the difficult points within the body where possible or in the appendix for future revisions. For example, we plan to include guiding examples for the neural network notation to clarify their meaning as well as further exposition for some of the subtle conceptual issues that arise.
>
> **Additional References:** Thank you for the great list of additional references. There is definitely some overlap and we plan to incorporate some of these in future revisions.

---

### Official Review · Reviewer_4f3Q · 2021-07-15

**Rating:** 6
**Confidence:** 2

**Summary:**

This paper studies the theoretical expressive power of the PointNet and (modified) DeepSets methods, on which many recent point-cloud neural-network methods are based. It proves both universality theorem under specific settings, and shows some of its theoretical limitations.


**Limitations And Societal Impact:**

The limitations are addressed in the paper. The societal impact is irrelevant.


**Main Review:**

The paper clarity could be improved. The abstract could be re-arranged to make the paper's contributions more obvious. In all, I found the paper hard to follow.

The experiment used is an excellent way of showing the error bound from Lemma 4.3

Minor remarks:
1. Line 43 - "different" used twice.
2. Line 213 - the beginning of the sentence is unclear.
3. It will help if it will be stated for each theorem, if its proof is in the appendix.

**Time Spent Reviewing:**

5

---

> ### Author Response · Authors · 2021-08-10
> **Thank you for your review and constructive feedback. Below we shall address the raised points.**
>
> **Clarifying Line 213:** What we write in Line 213 is to remind the reader of the meaning of the main theorem in [5], but in terms of the context and notation of this paper. The set $\mathcal N^\sigma$ is the set of all single hidden-layer neural networks with linear output layer and activation $\sigma$ (as introduced in the preceding section on Line 193), and we assume $\sigma\in\mathcal C(R)$ i.e. that the activation is continuous. Then the content of [5] is that this family of neural networks can uniformly approximate the continuous functions on a compact region of Euclidean space if and only if the activation $\sigma$ is *not* a polynomial. Indeed, if $\sigma$ were a polynomial, of say degree 10, then the neural networks under consideration could only represent polynomials which are of degree 10 or lower, which is not enough to approximate all continuous functions.
>
> **Clarifying Proof Location and Abstract:** The suggestion to explicitly state the location of the proof is a good one and will be incorporated. We will also improve the abstract in any future revisions to emphasize the paper’s contributions.
>
> **Typographical Errors:** We have noted the error and others and will make amendments for future versions.

---

> > ### Comment · Reviewer_4f3Q · 2021-08-24
> > **Response to the rebuttal**
> >
> > After reading the other reviewers responses, I became more aware to the contributions of the paper, and I support its acceptance. I believe that improving the paper's clarity will help much for future readers. Like another reviewer wrote - I would have stopped reading the paper early if it was merely for my own research purposes.
> > I'd like to increase my score to 6.

---

### Official Review · Reviewer_K3ic · 2021-07-16

**Rating:** 8
**Confidence:** 3

**Summary:**

This paper presents theoretical results on two prominent architectures for representing functions on sets, PointNet and (a normalised version of) Deep Sets (or equivalently, max-pooled functions and mean-pooled functions). The authors analyse these architectures from the perspective of appropriately-defined metrics on sets of sets, and the associated function spaces. They obtain universal approximation results which show that each architecture is capable of approximating all functions which are uniformly continuous with respect to the appropriate metric. There are also interesting results on the Lipschitz properties of the architectures and on the limitations of PointNet in particular.

**Limitations And Societal Impact:**

This work is very mathematically focused and I don't believe it has any potential negative societal impact (except perhaps very indirectly).

**Main Review:**

This paper is clearly a valuable, interesting and novel contribution to the theory of models for learning functions on sets. The architectures considered in this paper are of interest to a reasonably wide audience, and the results presented are novel, strong, and say interesting things about the approximation strength of these architectures. I was not able to check the details of the proofs for these results - my review assumes that the proofs are actually correct. I have seen nothing to make me doubt the mathematical competence of the authors.

In general the writing is clear and the paper is well-structured. The paper would benefit from a pass to check for grammar/punctuation/typos (e.g. the word "different" appears as "different different" on lines 42-43). The mathematics is necessarily densely presented, and would benefit from having more room for discussion/explanation - the page limit obviously limits what can be done here, but perhaps the experimental section could be moved to supplementary material to make room? I didn't find the experimental demonstration of Lemma 4.3 necessary to the narrative (though it is certainly an interesting illustration). Nevertheless, I don't have a strong opinion that this needs changing, and the mathematics was presented well given the tight constraints on space.

A minor point on related work: in the introduction subsection Cardinality Limitations, the authors note a lack of previous work on variable-cardinality sets. This is partially addressed (strictly speaking in the case of multisets rather than sets) in On the Limitations of Representing Functions on Sets (Wagstaff et al 2019), which the authors cite elsewhere in this paper but don't mention at this point in the paper.

**Time Spent Reviewing:**

4

---

> ### Author Response · Authors · 2021-08-10
> **Thank you for your review and constructive feedback. Below we shall discuss some of the raised points.**
>
> **Moving the Experimental Result:** Though we also agree that the experimental result is not necessary to the narrative as we believe the theoretical guarantee stands on its own, we hope the empirical verification helps to address any lingering doubt the audience has as to the correctness of the bounds. As a result, we will likely keep it in the main body. However, to make room for clarifying exposition in other areas, we can compress the exposition for the experimental section or move some details to the supplementary materials.
>
> **Cardinality Limitations:** Thank you for pointing out the relevance of Wagstaff et al 2019 to the Cardinality Limitation discussion. We will elaborate on the relationship of this work to ours in future revisions.
>
> **Typographical Errors:** We have noted the error and others and will make amendments for future versions.

---

> > ### Comment · Reviewer_K3ic · 2021-08-19
> > **Response to rebuttal**
> >
> > Thanks for your response. I take your point that some readers will find the experimental section necessary, and I think your suggestion of compressing this section rather than removing it entirely makes sense. My rating for this paper is unchanged.

---

### Official Review · Reviewer_GPE3 · 2021-07-21

**Rating:** 6
**Confidence:** 2

**Summary:**

This paper study the expressiveness of popular two processing models such as deepset and pointnet. This paper proves that (1) PointNet can uniformly approximate only the functions which are uniformly continuous with respect to the Hausdorff (Wasserstein) metric and (2) Only the constant functions can be simultaneously uniformly approximated by both PointNet and normalized-DeepSets when input sets are allowed to be arbitrarily large. A lower bound of PointNet to approximate the center-of-mass is proved and a toy experiment is performed to confirm this lower bound.


**Main Review:**

This is a good theoretical paper that investigates the representation power of set models when the cardinality becomes arbitrarily large. However, I am more interested in what benefit this paper can bring to practical applications.  The lower bound proved in this paper is only for center-of-mass, but in practical applications of the set, the tasks are not limited to estimate the mass center. Practitioners are also doing set compression, point cloud segmentation, classification, etc. Therefore I'd like to know how this paper could lead to the advancement of these application areas.

Also, Set Transformers [1] is the STOA model for set modeling. Can the theory introduced in this paper be applied to set transformers?

[1] Set transformer: A framework for attention-based permutation-invariant neural networks, ICML, 2019

**Time Spent Reviewing:**

2 hours

---

> ### Author Response · Authors · 2021-08-10
> **Thank you for your review and feedback. Below we shall address some of the raised points.**
>
> **The Error Lower Bound and Practical Implications:** First, we would like to point out that the error lower bound holds beyond just the center-of-mass and actually applies to all functions $ave_f$ where $f$ is continuous (see Theorem 4.2 & Lemma 4.3). The center-of-mass function is merely the simplest case ($f$=identity map). By choosing $f$ to be the monomials (e.g. $x^2$), these results also imply that PointNet can not uniformly approximate moments either. If the issue lied solely with the center-of-mass, one could presumably pre-center the data before using PointNet as a way to circumvent the unlearnability of this function. However, since the issue lies with average-pooling of any continuous function, it is unlikely that preprocessing can circumvent this issue completely. This is certainly something to emphasize and make clearer in future revisions.
>
> **Other Tasks:** Since this work provides negative results on function approximation for sets, it immediately has implications on regression over sets (as illustrated by the experiment). Hence, classification is also a casualty as such tasks are typically accomplished by appending a softmax layer at the output layer (mentioned briefly on Line 35) . For other tasks (e.g. set compression, set segmentation, etc) one usually still needs a permutation-invariant network for some stage, e.g. as a submodule, to produce useful features for the task and to ensure permutation-invariance. If such projects use the discussed set-pooling networks, then the results in this work will have relevance.
>
> **Set Transformer:** At the moment we are not certain how we could apply these techniques to set transformers or attention mechanisms more broadly, though this is definitely worthy of further investigation. The first step would be to identify the natural topology for their input space, but from there the particulars of the architecture will likely matter significantly.

---

> > ### Comment · Reviewer_GPE3 · 2021-08-20
> > **response to the rebutall**
> >
> > The first rebuttal "The Error Lower Bound and Practical Implications" addresses my concerns and I'd like to increase my score to 6.

---

### Official Review · Reviewer_prSH · 2021-07-30

**Rating:** 6
**Confidence:** 2

**Summary:**

This paper develops universal approximation theory for deep sets / pointNet architectures which consist of the composition of a deep network on a permutation invariant aggregation of embeddings (i.e. $f(x) = \rho \circ \text{agg} \circ \phi \circ x$ where 'agg' is sum(), max() or mean()). It was already known from the original Deep Sets paper [Zaheer et al 2017] that these architectures are universal approximators of set functions on finite and countably infinite input sets, and there exist negative results [Wagstaff et al 2019] for the uncountable case. This paper refines the story in the uncountable case: it shows that the choice of aggregation function (max or mean) defines the class of functions that can be approximated in the uncountable case, and they give lower bounds on approximation power in the finite input case.

**Limitations And Societal Impact:**

The authors are clear about the limitations of their work.

**Main Review:**

[Clarity] I found this paper frustrating to review because it's clear that a lot of careful work has gone into proving the results, but it is written in a way that makes their results almost impenetrable to those without a strong topology background. In many respects, I'm the perfect audience for this paper: I know the related work well, and am very interested in the approximation power of these methods, but honestly - if I was just reading this paper as a researcher (rather than a reviewer), I would have stopped at the end of page 3 (it is very clearly written up to that point). Now, that is a style choice that the authors are free to make, but I think it severely limits the potential audience of this work, and it's a shame because I think it could have been written in a way that teaches those readers who are unfamiliar with topology about how we can leverage existing results from topology in building approximation theorems. The prose in section 3.4 (which explains why it's hard to get the same results with sum pooling) is a good example of well-written, accessible text that I would have liked to see at the beginning of section 2 and section 3.

[Quality / significance] My main takeaway from this paper was that Deep Sets (with average pooling instead of sum pooling) and PointNet can each uniformly approximate different classes of functions (in the uncountable case) --- so the aggregation function matters --- and this difference in approximation power remains for finite sized sets (in the case of PointNet). From the counter example functions (centre of mass & diameter of a point cloud) it's intuitive that max / mean (respectively) would be the wrong choice of aggregation function, but I was not left with a good sense of what the properties of the two classes of functions are in general. I.e., given some new problem do the properties of the respective function classes give some guidance on the choice of aggregation function (as the paper suggests when it alludes to inductive biases)?

I also didn't get a good sense of the relationship between these two classes of functions. For finite sized inputs, we can smoothly vary between the two aggregation functions using a weighted sum, $\sum_{i} \alpha(x_i / T) x_i$, where the $ \alpha(x_i / T)$ is the output of the softmax function with temperature $T$ (mean and max are the limits as $T$ goes to 0 and $\infty$ respectively) --- are there analogous  weighted sums that allow us to approximate both classes of functions in the infinite case? (I am tempted to ask analogous questions about what attention-weighting schemes buy us, but it's not clear how the attention weights are even defined if you have infinitely large sets - feel free to comment if you have thoughts).

Overall, I thought this was a useful contribution that was let down by it's presentation. I don't have the background to evaluate the correctness of the theoretical claims, but assuming that they are correct, I think that it deserves to be published. Still - I would strongly encourage the authors to try make the paper more broadly accessible.

**Time Spent Reviewing:**

8

---

> ### Author Response · Authors · 2021-08-10
> **Thank you for your review and constructive feedback. Below we will address the points you brought up.**
>
> **Prose:** Thank you for the constructive feedback on this. Unfortunately, we had difficulty given the space constraints in rigorously describing the results while keeping things accessible, but there are certainly places where improvements can and should be made. For example, motivating some of the topological tools and expanding upon the neural network notation with examples and exposition would certainly be beneficial.
>
> **Relationship between the Two Function Classes:** Surprisingly, the situation for unbounded finite cardinality and bounded cardinality are radically different. For unbounded finite cardinality, the two functions spaces only agree on constants for topological reasons (Theorem 4.1). For bounded cardinality, there is much greater overlap, in fact, by the results of Deep Sets [16], DeepSets can approximate at least as many functions as PointNet for fixed cardinality, and by the error lower bound (Theorem 4.2 & 4.3) we know that PointNet can’t approximate certain functions that DeepSets can. This seemingly paradoxical situation is due to the fact that it is possible for set cardinality to go to infinity and nevertheless converge to a finite set. This is illustrated in the proof of Theorem 4.1 in the appendix (line 583 contains the crux of the matter). This is a subtle topic which we should certainly expound upon in any future revisions.
>
> **Theory-Guided Aggregation & Inductive Bias:** In the bounded cardinality case (which is the case of most practical interest), the error lower bound shows that all functions of the form $ave_f$ where $f$ is continuous cannot be uniformly approximated by PointNet. As a special case (which we should mention in future revisions) this means PointNet cannot uniformly approximate moments (choose $f$ to be a monomial e.g. $x^2$). This suggests that if the learning tasks depends on the *distribution* of the sample points and not merely the *shape* of the samples, then max-pooling is not the way to go, and one should opt for average pooling (or sum-pooling). On the other hand, if domain knowledge suggests that the task should only depend on the shape of the point cloud, and not the distribution of the points (e.g. it shouldn’t matter whether the poles of a sphere are oversampled) then this inductive bias can be implemented by choosing max-pooling over average/sum-pooling.
>
> **Interpolating Pooling Functions:** This is a great question. It is not immediately clear what the natural topology that arises from softmax-interpolation would be or even from the more mundane linear interpolation t*max+(1-t)*ave. And without a topology, much of the tools developed in the paper are inapplicable. The resulting natural intermediate topologies feel like they should lead to either a new function class which is somehow “inbetween” or would be the function class associated to the “stronger” pooling function (average-pooling) in bounded cardinality case. However, all this is just speculation at this stage.
>
> **Attention Weighting Schemes:** Unfortunately, we do not currently see an immediate way to apply this to attention-weighting schemes for the general cardinality case, but a first step would be to identify a natural topology on the input space for such networks and then work from there.

---

### Official Review · Reviewer_U3mP · 2021-08-03

**Rating:** 6
**Confidence:** 4

**Summary:**

The authors prove universality results for permutation invariant networks: in particular for PointNet and 'normalized DeepSets'. The two classes of networks differ in the pooling layer: max pooling for PointNet and averaging for Normalized DeepSets.

Despite their apparent similarity, the authors show that these two networks approximate a different set of functions and according to different topologies (Haussdorf vs Wasserstein).
A second important contribution is that the paper shows uniform approximation results while allowing for varying number of inputs which can be potentially uncountable (Thm 3.4). This is novel compared to recent prior work which considered fixed input cardinality. It also requires different mathematical tools for achieving this.

A third contribution comes in the form of thm 3.7 which shows these networks to be Lipschitz. Along with Thm 3.4, this allows to control the approximation error made by the network when the input signal is subsampled.

Finally, the authors identify a failure case where PointNet fails to approximate a seemingly simple function 'average' (Thm 4.2 and Lemma 4.3), thus illustrating the theory from which it is reach the same conclusion as the failure case does not belong to the set of functions that can be approximated by PointNet.

--------------------------------------------------------------------------

Update: After reading all other reviews and the authors' responses, I think still think the paper might be of interest to the community and is thus worth publishing under the condition that the clarifications asked by the reviewers are taken into account.



**Limitations And Societal Impact:**

Yes

**Main Review:**

General comments:
- Significance: This is a significant work as it provides a clear distinction between the approximating power of two popular set networks : PointNet and DeepSets. The results have practical consequences which can be useful to the ML community for choosing the most appropriate network architecture depending on the task.
- Originality:  The work is original in that it treats both classes of networks using the same theoretical framework while still capturing meaningful qualitative differences between the two classes.
- Quality: The proof technic relies on standard Stone-Weierstrass theorem as well as the UAT [6] and do not represent a significant challenge, but the results themselves are useful.
- Clarity: This is the weakest point of the paper. The notations are often cluttered and heavy especially in section 2.

Detailed comments:
Notations:
-Notations are defined at several places in section 2, which makes it difficult to localize where they are defined especially in a paper with heavy notations.
-The notations seems unnecessarily complicated for a rather simple concept in particular in section 2.4. The authors should simplify or clarify and illustrate these notations using examples.
For instance, it is hard to parse the nuances between the different notations for sets of networks as they only differ in a particular subscript or superscript : $\mathcal{N}^{\sigma}$, $\mathcal{N}^{\sigma,\tau}$, $\mathcal{N}^{\sigma,\tau}_{Max}$, $\mathcal{N}^{\sigma,\tau}_{max}$, etc.

Definitions:
The authors discuss two ways to build deeper networks either inductively or using 'deeper weights' (l:199), however, it is hard to relate these constructions to the notation that is introduced. In particular, the authors use the notation $\mathcal{N}^{\sigma,\tau}$ where sigma and tau are either nonlinearities or a collection of nonlinearities (when written in bold). However, in the latter case (collection of nonlinearities), it not clear how the set of functions $\mathcal{N}^{\sigma,\tau}$ is defined.


Illustrative figures:
The authors should illustrate the importance of having an approximation result that is independent on the input dimension. While the authors mention the case of subsampling a signal at different resolutions, it might be beneficial to illustrate this on simple visual examples. In particular, this could give a better intuition for failure cases in section 4.



Discussion of related work:
As the authors point out, Thm 3.1 is a general known result. Can the authors explain further why the result of [11] cannot be applied for PointNet but only to normalized DeepSets?


Approximation power of PointNet:

The phrasing in l 324 suggests that PointNet is less powerful than normalized DeepSet, however, as discussed earlier in the paper, they just approximate different sets of functions (that are not really comparable) using different topologies. In that sense, thm 4.2 which quantifies the error made by a PointNet in approximating an average $ave_f$ might be misleading. Wouldn't it be possible to obtain worst case errors for normalized deepSet using a suitable function like max_f(A)?

Approximation power of standard DeepSets:
On section 3.4 : The authors mention that standard DeepSets are not covered by the proposed framework and provide a nice explanation for why this is the case in general. Could it be possible to express a standard DeepSets as a normalized one when the nonlinearity sigma is positively homogeneous (like ReLU)? In this case, it should be possible to absorb the normalizing constant in the weights of the second network.


**Time Spent Reviewing:**

5

---

> ### Author Response · Authors · 2021-08-10
> **Thank you for your detailed review, constructive feedback, and thoughtful questions. Below we shall address the key points in your review.**
>
> **Notation & Definitions:** The neural network notation in the paper was intended to provide precision, generality, conciseness, and utility for the proofs. Regrettably, it seems that some clarity was lost along the way. As a remedy, Section 2.4 can be elaborated upon via discussion of the notation and by providing concrete examples of the functions that live within these neural network classes. Alternatively, with extra work, it may be possible to refactor the paper and results so that the notation is relegated to the appendix as a technical tool with only the consequences appearing in the main body in plainer language. We would be happy to consider any of these two options for any future revision.
>
> **Clarification of Inductive Construction of Deeper Networks:** There is a nontrivial difference between semicolons “;” and commas “,” when used as separators of nonlinearities in the neural network notation. The comma just means the nonlinearities occur sequentially. So if one sees $...,\sigma, \tau,...$ this means $\sigma$ is the nonlinearity for one layer and $\tau$ is the nonlinearity for the next layer. Writing a vector nonlinearity just means there is a sequence of nonlinearities, i.e. if $\sigma=(\sigma_1,...,\sigma_9)$ then there is a layer with $\sigma_1$ nonlinearity, the next layer with $\sigma_2$ nonlinearity, the next layer with $\sigma_3$ nonlinearity, and so on. A semicolon has a different purpose and it means that the two nonlinearities are separated by a set-pooling operation which is denoted in the subscript (in the manner given by the constructions in Section 2.4). The left side of the semicolon corresponds to the nonlinearities for the “inner-function” of PointNet/DeepSets and the right side of the semicolon corresponds to the nonlinearities of the “outer-function”.  Thus, in addition to just talking about the depth, we can talk about how the depth is distributed between the outer/inner-networks. Lastly, the subscripts serve two purposes: (1) to remind the reader whether or not there is a set-pooling operation involved and (2) to remind the reader of the input space. When there is no subscript, the domain of the networks is Euclidean space (these are traditional neural networks). When the subscript is max/ave/sum then this should remind the reader the domain is now a space of finite subsets. When the subscript is Max/Ave, this is to remind the reader that we are now working in the extension spaces and that these networks have domain $\mathcal K(X)$ or $\mathcal P(X)$.
>
> **Illustrative Figures:** It is unfortunately difficult to directly illustrate the approximation results on $Fin(X)$ since this space is in general infinite dimensional. Similarly, $Fin^m(X)$ has intrinsic dimension $m*dim(X)$ which is generally very high. However, it may be possible to illustrate certain aspects of this for very low cardinality/dimension combinations, and this is certainly worth looking into. Alternatively, we have prepared some illustrations to help clarify some of the mathematical concepts involved in the paper.
>
> **Stinchcombe’s Theorems and PointNet:** As mentioned on Line 217, Theorem 3.1 and Stinchcombe’s Theorem 5.1 in [11] are in essence implying the same thing. And so anywhere Theorem 3.1 is used, with a little translation, Stinchcombe’s Theorem 5.1 can also be used. The inclusion of our alternate but essentially equivalent result is so as to be self-contained, provide a different proof, and to side-step the additional translation overhead needed to use the result (as Stinchcombe creates specialized and technical notation motivated by his use case). Additionally, we mention on Line 248 that Stinchcombe’s Corollary 5.1.2 in [11] can be applied to normalized-DeepSets. This is because our space of probability measures $\mathcal P(X)$ is a compact subset of the space of signed measures $\mathcal M(X)$ (with weak-* topology) which is itself a locally convex vector space with topological dual space $\mathcal M(X)^*\simeq\mathcal C(X)$. The functions $Ave_f$ can be thought of as linear functionals on the space $\mathcal M(X)$ arising from integration with against members $f\in\mathcal C(X)$ (the famous Riesz-Markov representation theorem, briefly mentioned on Line 167). Lemma 3.2 ensures that even when $f$ are restricted to be neural networks, the family of functions $Ave_f$ has the right conditions for Theorem 3.1 and as a consequence Stinchcombe’s Corollary 5.1.2. To apply Stinchcombe’s corollary to PointNet, we would first need to embed the Hausdorff metric space $\mathcal K(X)$ into a locally convex vector space in which the functions $Max_f$ can be thought of as a sufficiently rich family of linear functionals with all the the right properties and compatibility conditions. Assuming all this can be made to work, we would not consider this a “direct application” as in the case of normalized-DeepSets, where things work out nearly immediately.
>
> **Approximation Power of PointNet:** Your interpretation of Line 324 is correct and the confusion has to do with the radically different nature of approximation with unbounded cardinality and bounded cardinality. For unbounded finite cardinality (the space $Fin(X)$), the two functions spaces only agree on constants for topological reasons (Theorem 4.1). For bounded cardinality, there is much greater overlap, in fact, by Theorem 9 of Deep Sets [16], we know DeepSets can approximate at least as many functions as PointNet for fixed cardinality, but by the error lower bounds (Theorem 4.2 & 4.3) we know that PointNet can’t approximate certain functions that DeepSets can. This seemingly paradoxical situation is due to the fact that it is possible for set cardinality to go to infinity and nevertheless converge to a finite set. In a very informal sense, it’s as if the “cardinality-axis” of $Fin(X)$ (partially) wraps around so that you can go off to “infinity” and return to the finite sets. However, the way things “wrap-around” is fundamentally different for the two metric topologies and hence yield radically different function spaces when unbounded cardinalities are considered. This is illustrated in the proof of Theorem 4.1 in the appendix (Line 583 contains the crux of the matter). If you truncate the cardinality to not allow the infinite cardinality limit, you can no longer wrap around, and things behave in a nicer manner where the approximable function classes are now comparable. This is a subtle topic which we should certainly expand upon in any future revisions.
>
> **Approximation Power of DeepSets:** As mentioned in the paragraph starting at Line 112, the internal normalization by cardinality indeed makes a difference, and we mention there how the difference manifests as cloud cardinality goes to infinity. However, in the specific case of fixed cardinality, the normalization factor becomes a constant and you are indeed correct that then the normalization factor can be absorbed. On the other hand, note that if the cardinality is allowed to change, the internal normalization factor is no longer a constant (it now depends on the input). As such in the variable-cardinality case it can’t be absorbed into the weights or nonlinearities. This is something we should elaborate on in future revisions.

---

> > ### Comment · Reviewer_U3mP · 2021-08-23
> > **The response addresses most of my concerns**
> >
> > Thank you for you response. I believe the proposed improvements to the notation by introducing a detailed and well explained version in the appendix and favor an easier to follow description in the main text can greatly improve the readability of the paper and make it accessible to a wider audience. In general, I believe it is better to use visually similar notations for similar concepts and very different notations for concepts that differ greatly in the context of the paper. This could apply to the semicolon ';' vs coma ',' example.
> >
> > Thank you for the clarifications about the remaining points, I think it might be useful to mention such discussions, especially for Stinchcombe’s thm and Approximation Power of PointNet, whenever space permits or in the appendix.

---

### Decision · Program_Chairs · 2021-09-27

**Decision:**

Accept (Poster)

**Comment:**

This paper gives a novel theoretical understanding on what functions "normalized" DeepSets and PointNets can represent, from a topological point of view. The work is novel and provides meaningful guidance to the community about the difference between these approaches; many of the (numerous) reviewers, however, had complaints about the clarity of the presentation as well as confusions on some specific points. It will greatly benefit the paper to take these into account for the camera-ready version.